# Pharmacology of Free Fatty Acid Receptors and Their Allosteric Modulators

**DOI:** 10.3390/ijms22041763

**Published:** 2021-02-10

**Authors:** Manuel Grundmann, Eckhard Bender, Jens Schamberger, Frank Eitner

**Affiliations:** 1Research and Early Development, Bayer Pharmaceuticals, Bayer AG, 42096 Wuppertal, Germany; frank.eitner@bayer.com; 2Drug Discovery Sciences, Bayer Pharmaceuticals, Bayer AG, 42096 Wuppertal, Germany; eckhard.bender@bayer.com (E.B.); jens.schamberger@bayer.com (J.S.)

**Keywords:** GPCR, allosteric modulator, Free fatty acid receptor, FFAR, drug discovery

## Abstract

The physiological function of free fatty acids (FFAs) has long been regarded as indirect in terms of their activities as educts and products in metabolic pathways. The observation that FFAs can also act as signaling molecules at FFA receptors (FFARs), a family of G protein-coupled receptors (GPCRs), has changed the understanding of the interplay of metabolites and host responses. Free fatty acids of different chain lengths and saturation statuses activate FFARs as endogenous agonists via binding at the orthosteric receptor site. After FFAR deorphanization, researchers from the pharmaceutical industry as well as academia have identified several ligands targeting allosteric sites of FFARs with the aim of developing drugs to treat various diseases such as metabolic, (auto)inflammatory, infectious, endocrinological, cardiovascular, and renal disorders. GPCRs are the largest group of transmembrane proteins and constitute the most successful drug targets in medical history. To leverage the rich biology of this target class, the drug industry seeks alternative approaches to address GPCR signaling. Allosteric GPCR ligands are recognized as attractive modalities because of their auspicious pharmacological profiles compared to orthosteric ligands. While the majority of marketed GPCR drugs interact exclusively with the orthosteric binding site, allosteric mechanisms in GPCR biology stay medically underexploited, with only several allosteric ligands currently approved. This review summarizes the current knowledge on the biology of FFAR1 (GPR40), FFAR2 (GPR43), FFAR3 (GPR41), FFAR4 (GPR120), and GPR84, including structural aspects of FFAR1, and discusses the molecular pharmacology of FFAR allosteric ligands as well as the opportunities and challenges in research from the perspective of drug discovery.

## 1. Introduction

### 1.1. Allosterism at G Protein-Coupled Receptors

G protein-coupled receptors (GPCRs) are the largest group of membrane proteins and play key roles in various physiological and pathological processes. The main function of GPCRs is to constantly monitor the cellular environment and signal to adapt the cellular responses to biological events over short and long distances throughout the body. GPCRs are targeted by endogenous and exogenous molecules via binding to specific sites at the receptor protein, stabilizing and/or inducing conformational changes that dictate the signaling properties of the receptor-adaptor molecule interaction and thus downstream signaling. The exact mechanisms of receptor activation and transduction of information are not fully understood and remain a matter of intense research. Known modulators that can substantially change the behavior and signaling of GPCRs in vivo are, for example, (i) (sub)cellular localization of the receptors, including spatiotemporal regulation of GPCR–effector interaction, (ii) homo- or heteromeric multimerization of receptors, (iii) graded ligand efficacy, e.g., full versus partial agonism, (iv) multidimensional efficacy, i.e., signaling bias, and (v) allosteric modulation. As the latter phenomenon is the focus of this review, allosterism and its most common shapes shall be introduced.

Allosteric modulation is a biological phenomenon not only restricted to GPCRs, but occurs in practically all biologically active molecules. The concept of allosterism is considered and widely used in drug discovery and development, with several molecules that act allosterically being clinically applied [1,2]. The essential mechanism is the transduction of information, e.g., in the form of molecular conformation or ensembles of conformations, from one location to another. If an endogenous ligand binds to one binding site, i.e., the orthosteric site, allosteric ligands can bind to a site spatially distinct to this orthosteric site—i.e., an allosteric site. It becomes clear that GPCRs harbor several potential binding sites for ligands, so that although by definition only one orthosteric site exists, several allosteric sites can theoretically be identified. The transduction of information by the interaction between orthosteric and allosteric ligands can come in various and at times complex shapes. Allosteric ligands can either increase, leave unchanged, or decrease the affinity and/or the efficacy of another ligand—e.g., orthosteric ligand. Additionally, a ligand that binds to a site distinct from the orthosteric site can exert effects independently of a second ligand. If the allosteric ligand creates intrinsic efficacy, one would consider it an allosteric agonist. In the case of a neutral or decreasing effect on efficacy, the ligand is denoted as a neutral allosteric antagonist or inverse allosteric agonist, respectively. In many cases, the nature of the interaction between allosteric and orthosteric ligands gives the ligand its name. An allosteric ligand that increases the affinity and/or efficacy of an orthosteric ligand is a positive allosteric modulator (PAM), while an allosteric ligand that decreases the affinity and/or efficacy of an orthosteric ligand is a negative allosteric modulator (NAM). The magnitude and direction of an interaction are continuous and independent characteristics of allosteric ligands. Thus, ligands also exist that do not modulate one or more parameters of an orthosteric ligand. Such a ligand can be called a silent allosteric modulator (SAM) or neutral allosteric ligand (NAL). Additionally, combinations are possible, e.g., PAM-antagonists, which increase the affinity of the orthosteric ligand but reduce the efficacy of the orthosteric ligand [3]. Ago-PAMs are agonists that are also allosteric modulators.

A characteristic feature of allosteric interaction is reciprocity, which is the effect of the allosteric ligand to the orthosteric ligand, which also applies the other way around—i.e., from the orthosteric ligand to the allosteric ligand. Another key feature of allosteric communication is probe dependency—i.e., that a given allosteric ligand might increase the affinity of a certain orthosteric ligand but could show a completely different effect when cobound with another orthosteric ligand [4]. Terms such as PAM or NAM thus describe a relationship rather than a single ligand, although they are most often used to describe certain molecules. Hence, a general classification can only fall short of detailed characteristics and cannot describe the ligand features holistically. 

The interdependent relationship between the ligands can also be described mathematically—for instance, based on the ternary complex model where the receptor is subjected to binding of both an orthosteric ligand and an allosteric ligand. The modulation of the affinities of the orthosteric and allosteric ligands to the receptor is described by the α factor. The modulation of efficacy of the activated receptor is described by the β-factor and is the ratio of the operational efficacy of the ternary complex (receptor + orthosteric + allosteric ligand) and the operational efficacy of the binary receptor [5]. Intrinsic efficacy of agonists, neutral or inverse agonists, can be described by τ. To ease the approach toward allosterism at GPCRs, some important classes of allosteric modulators are visually represented in Figure 1.

Homo- and heteromerization of two polypeptides (dimers) or higher order complexes (oligomers) of receptors are increasingly found for GPCRs, not only in artificially overexpressing cell systems but also in vivo [6]. This aspect of GPCR biology adds a further layer of complexity to the phenomenon of allosterism at GPCRs. In such a scenario, ligands binding to one protomer could cause conformational changes in the partner protomer via an intermolecular mechanism with effects on, for instance, affinity of ligands binding to the second protomer or G protein binding or on signaling events downstream of either protomer. GPCRs taking part in oligomers themselves function as allosteric modulators, as G proteins do in a simple monomeric GPCR model. Therefore, so called orthosteric ligands can turn out to be allosteric modulators when they convey allosteric effects from one protomer to another protomer that may or may not have a ligand bound to it. Allosteric modulation between homomeric GPCR dimers was early on reported for the dopamine D2 receptor [7] or the muscarinic acetylcholine M2 receptor [8]. Heteromeric GPCR complexes are increasingly thought to further shape GPCR signaling, such as adenosine A2a-A1 or A2a-D2 [9,10], D1-D3 dopamine [11] chemokine CXCR2, and δ-opioid receptors [12] or angiotensin II AT1-F2α prostanoid receptors [13], to give only few examples. The free fatty acid receptors FFAR2 and FFAR3 have also been found to build heterodimers in both recombinant and primary cell environments [14]. Ang et al. demonstrated that the FFAR2-FFAR3 dimer constitutes a bona fide allosteric system with the FFAR3 protomer functioning as a positive allosteric modulator of FFAR2 signaling (Ca^2+^ increase downstream of FFAR2) and FFAR2 allosterically modulating FFAR3 function vice versa (β-arrestin 2 recruitment to FFAR3). However, signaling from FFAR2-FFAR3 dimers did not only differ quantitatively but also showed different qualities, as the dimeric receptor signaling fingerprint lacked Gi/o-mediated cAMP decrease, a feature of both monomeric receptors, but gained the ability to induce p38 phosphorylation [14]. These data illustrate that the multiprotein architecture of GPCR oligomers provides plenty of opportunities for allosterism to occur.

### 1.2. G protein-Coupled Receptors Activated by Free Fatty Acids (FFARs)

Physiologically occurring medium- (MCFAs, 6–12 carbons) and long-chain fatty acids (LCFAs, >12 carbons) are mainly derived from dietary fat intake or metabolic turnover of triglycerides. Short-chain fatty acids (SCFAs, <6 carbons) are produced by bacterial fermentation of dietary fibers or, to a lesser extent, originate from fermented food products. The biological activity of free fatty acids has long been associated with only their function as energy substrates and their ability to engage in metabolic pathways. Around the beginning of the new millennium, researchers found that membrane receptors could sense products of intermediate metabolisms to regulate host responses toward changing metabolic environments. Five years after the discovery of genes encoding for GPR40, GPR41, and GPR43 in 1997 [15], GPR40 was deorphanized. The receptor was found to respond to medium- and long-chain fatty acids (saturated and unsaturated) as well as thiazolodinedione drugs and its function on pancreatic β-cells was outlined [16,17,18]. GPR40 was subsequently renamed FFAR1. At the same time, GPR41 and GPR43 were also deorphanized by discovering short-chain fatty acids as their endogenous ligands and were thus renamed FFAR3 and FFAR2, respectively [19,20,21]. Another 2 years later, another orphan GPCR, GPR120, was discovered [22] and was identified as responding to long-chain fatty acids and as having an ability to promoted glucagon-like peptide 1 (GLP-1) secretion [23] and subsequently renamed to FFAR4. The last receptor to be covered in this review is GPR84, which was discovered in 2001 [24] as a GPCR expressed on leukocytes [25] and was later found to be activated by medium-chain fatty acids [26]. However, GPR84 is still deemed to be an orphan receptor and awaits official deorphanization status, although some researchers discuss it as a putative future FFA5 receptor [27]. 

The signal transduction pathways triggered by the free fatty acid receptors span the whole repertoire of G proteins and noncanonical effector molecules such as β-arrestins. While the FFA1 receptor couples to members of all major G protein families, i.e., Gi/o, Gq/11, Gs, and G12/13, and additionally recruits β-arrestin upon activation, so far, only Gi/o, Gq/11, and β-arrestin recruitment have been linked to FFA2 receptor pharmacology, whereas FFA3 receptor signaling seems to be only channeled via Gi/o proteins, although β-arrestin recruitment was reported as well [14]. Although the FFA4 receptor has been linked to both Gq/11 and Gi/o in a physiological context [28,29], the short isoform of the receptor can signal only via Gq/11 in a heterologously expressing system, but also recruits β-arrestin after activation, whereas the long isoform, harboring a 16 amino acid sequence in the third intracellular loop compared to the short isoform, does not induce calcium signaling but engages with β-arrestin upon activation [30]. Because putative G protein-independent, “stand-alone” β-arrestin signaling downstream of GPCRs lacks valid scientific evidence [31] and given that whole cell response toward activation of long isoform FFA4 receptors, although largely decreased, was still detectable using a pathway-unbiased holistic biosensor approach reported by Watterson et al. [30,32], it is tempting to speculate that it is actually remaining G protein activity downstream of FFAR4 that underlies this bias phenomenon of the short receptor isoform. However, solid experimental efforts are needed to prove this hypothesis. Finally, GPR84 has been described to couple with β-arrestin and Gi/o but not G12/13, Gq/11 or Gs family proteins [26,33,34]. A summary of the FFA receptor expression profile and biological function is shown in Table 1.

## 2. Free Fatty Acid Receptor 1 (FFAR1) 

### 2.1. Introduction

In accordance with the deorphanization of FFAR1 as a long-chain fatty acid sensing receptor, focus was put on the pancreatic expression of FFAR1, especially on β-cells, and its involvement in the secretion mechanisms of insulin [16,18]. Over the years, the expression pattern has been uncovered to be much broader, ranging from brain cells to bone and monocytes, and intestinal cells to rodent taste buds. Due to their shared ligand space, interpreting studies using fatty acids as physiological stimuli to deduce specific FFA receptor functions is not straight forward. Discovery of selective and potent (synthetic) agonists or antagonists as well as genetically engineered systems have helped to gain greater insight into the function of each of the receptors in particular tissues. For this reason, only studies using either genetically modified systems or chemically selective compounds targeting FFAR1 have been considered in the following.

### 2.2. Pancreatic β-Cell Function

In the description of the biological function of the FFA1 receptor, the focus is on the regulation of metabolic hormone secretion—that is, insulin secretion by pancreatic β-cells and gut hormones (GLP-1, GIP, and CCK) by enteroendocrine cells (see below). Various mechanisms have been discussed and experimentally examined based on the observation that fatty acids, especially long-chain saturated and unsaturated fatty acids, affect metabolic homeostasis. Several studies found that fatty acids enhance glucose-stimulated insulin secretion (GSIS) via the FFA1 receptor by using receptor-deficient or knock-down models (reviewed in [35]). Long-term treatment with fatty acids, however, decreased glucose tolerance and led to detrimental effects on the pancreatic β-cell, a phenomenon known as lipotoxicity [36]. In the early years of FFAR1 research, discussions arose as to whether FFAR1 might be responsible for both the insulin increasing as well as the lipotoxicity effects. It was reported that FFAR1-mediated acute insulin increasing effects coincide with chronic lipotoxic effects, such as impaired glucose homeostasis and hypoinsulinemia, with FFA1 receptor-deficient mice being protected against obesity-induced liver steatosis, hyperglycemia, glucose intolerance, and hyperinsulinemia [37]. Further studies reinforced this finding: one showed that chronic stimulation of human islets with palmitate ex vivo leads to decreased GSIS, which is further exaggerated by FFAR1 stimulation with the selective FFAR1 partial allosteric agonist TAK-875, the first clinically exploited allosteric modulator of the entire FFAR family, and reduced by FFAR1 inhibition with ANT203 [38]; another study demonstrated the beneficial effects of treating type 2 diabetic db/db mice with the FFAR1 antagonist DC260126 in terms of hyperinsulinemia and insulin sensitivity [39]. In contrast, the majority of available studies report that the detrimental chronic FFA-mediated effects are either FFA1 receptor-independent or FFAR1 is even protective against lipotoxic effects [40,41,42,43]. Other studies could show that FFAR1 is only one mediator of fatty acid effects on metabolic hormone homeostasis and apparently does not account for the lipotoxic effect of long-term fatty acid administration [40]. FFAR1-deficient mice were also not protected against high-fat diet-induced metabolic dysfunction assessed in glucose and insulin tolerance tests and liver fat accumulation [42]. Furthermore, FFAR1 activation with the selective orthosteric agonist TUG-469 was shown to be protective against palmitate-induced β-cell apoptosis [44], a finding that could be further refined by indicating that MAPK pathway activation downstream of FFAR1 is responsible for this protective effect [45]. However, whether signaling bias is truly a distinctive feature of FFA1 receptor involvement in the phenomenon of lipotoxicity needs further research.

While it became clear that elevated plasma free fatty acid levels, e.g., by dietary ingestion, could enhance glucose-stimulated insulin secretion (GSIS) from pancreatic β-cells, it remains unknown under which specific physiological or dietary conditions FFA1 receptors come into play. While around 50% of FFA-mediated GSIS enhancement was mediated via FFA1 receptors, the remaining part essentially relied on the metabolism of fatty acids [40]. Plasma fatty acids are almost completely bound to carrier proteins and the resulting fraction of free unbound fatty acids available to activate FFA1 receptors is considerably low [46]. In addition, the concept that dietary fatty acids could serve as the endogenous FFAR1 agonists and account for enhance glucose-mediated insulin secretion is challenging to prove given the fact that while postprandial glucose level increases, plasma FFA level decreases after meal ingestion. Another view on the FFA1 receptor was presented by Tunaru et al., who found that pancreatic β-cells produce the potent FFA1 agonist 20-Hydroxyeicosatetraenoic acid (20-HETE) upon glucose stimulation and proposed a novel autocrine feed-forward mechanism of endogenous FFA1 receptor activation in β-cells during GSIS. This mechanism seems to be impaired under type 2 diabetes conditions, as glucose-mediated 20-HETE production was decreased and both a 20-HETE formation blockade as well as FFAR1 antagonists were largely ineffective in blocking GSIS under type 2 diabetes [47]. 

### 2.3. Enteroendocrine Function

In addition to the pancreas, intestinal FFAR1 expression has been another focus of investigations around the biology of this receptor. FFA1 receptors can be found on several enteroendocrine cells, including L, K, and I cells, and are associated with glucagon-like peptide 1 (GLP-1), gastric-inhibitory peptide (GIP), and cholecystokinin (CCK) secretion, respectively [48,49]. Edfalk et al. showed that dietary fatty acids mediated GLP-1 and GIP secretion via the FFA1 receptor [48] and Liou et al. demonstrated that olive oil ingestions in FFAR1 expressing mice, but not in FFAR1-deficient mice, led to plasma CCK increase [49]. Interestingly, experiments using isolated perfused intestines from rats suggest that vascular but not luminal administration of endogenous and synthetic FFAR1 agonists can induce GLP-1 secretion [50]. Whether this is due to differences in apical versus basal receptor localization and associated signaling needs further elucidation. Another study followed up on this and found that chylomicrons were able to induce GLP-1 and GIP secretion from mouse and human intestinal tissues and a human peptide-secreting cell line model, suggesting that the effects of dietary fatty acids on incretin secretion found in vivo/in vitro and ex vivo could rely on their systemic or basolateral activities after absorption by the intestine and repackaging as triglycerides into chylomicrons and basolateral release [51]. The involvement of FFAR1 in this process is not entirely clear as the FFA1 receptor antagonist GW1100 could block cell line-derived incretin secretion but not murine tissue-derived incretin [51].

Additionally, the signal transduction composition downstream of FFAR1 appears to determine whether sole GSIS activity via β-cells or combined pancreatic GSIS and incretin secretion via enteroendocrine cells occur. Hauge et al. showed that agonists that induce both Gs and Gq signaling evoked GLP-1 and GIP release more effectively than agonists that induce Gq signals only, such as endogenous orthosteric ligands [52]. However, due to tissue- and cell type-dependent differences in receptor reserves as well as G protein abundancy and preference in different cell types, it is not clear whether it is actually the induction of different receptor conformations that leads to distinct signaling outcomes (i.e., signaling bias), or the graded strength of receptor activation—i.e., full versus partial agonism. 

### 2.4. Nervous System 

For brain FFAR1 localization, most studies describe associations of FFAR1 expression, fatty acid intake or FFAR1/FFAR4 dual agonist application with neurological phenotypes. Aizawa et al. compared FFAR1-deficient mice with wild-type mice in several behavioral tests and suggested an involvement of FFAR1 in both anxiety and depression behavioral traits [53]. The same group also investigated the role of FFAR1 in pain perception. They found that both mice receiving the FFA1 receptor antagonist GW1100 as well as those lacking FFA1 receptors were more sensitive to incision-induced mechanical allodynia, while FFAR1 knock-out had no effect in the plantar test that assesses thermal hyperalgesia [54]. Another report suggests that FFAR1 activation could be associated with antinociception as intrathecal injection of GW9508, a dual FFAR1/FFAR4 agonist, decreased mechanical allodynia and thermal hyperalgesia in neuropathic rats [55]. Future studies will have to show whether pharmacological intervention with this receptor harbors therapeutic value or whether existing agonistic treatments affect the nervous system in this regard. 

### 2.5. Bone Function

Other reports suggest an additional role of FFAR1 in bone, as it is expressed on several bone cells, including osteoclasts, osteoblasts, and osteocytes [56]. While RNA-mediated receptor knock-down experiments suggested that the FFA1 receptor mediates osteocyte apoptotic signaling of thiazolidinedione application, another study supports the notion that FFAR1 activation is protective against bone loss using FFAR1 knock-out mice together with the FFAR1/FFAR4 dual agonist GW9508 [57].

### 2.6. Molecular Receptor Pharmacology and Drug Discovery Efforts 

The pharmacology of the FFAR1 molecular receptor is complex and not fully understood. The receptor is reported to couple to all four heterotrimeric G protein families (Gq, Gi, Gs, and G12) as well as to β-arrestins [58,59]. It is tempting to speculate that the fully available repertoire is differentially covered by various ligands, resulting in a phenomenon known as signaling bias, where certain pathways are preferentially activated over others. The concept of signaling bias carries the potential to open new dimensions in therapeutic efficacy and selectivity. However, the translation from in vitro to in vivo models across different cellular systems and species has been shown to be challenging in general drug discovery approaches and even more so for molecules that exert differential signaling, such as biased ligands. One needs to bear in mind that studying signaling bias is complicated by the fact that this phenomenon is highly dependent on the cellular context and terms, such as partial or full agonist, as well as the biased ligand being only relative—e.g., a partial agonist for a certain pathway in a certain tissue background might appear to be a full agonist in this pathway in another tissue harboring another composition or equipment of signaling molecules. It becomes clear that coining ligands as functionally selective or generally full or partial agonists in terms of their efficacies falls short and does not reflect the biological reality adequately because efficacy depends on the biological context and is not a fixed parameter. Nevertheless, for the sake of simplicity and providing an overarching understanding of the complex processes at the receptor, the different ligands described in this review shall be grouped according to observed signaling engagements in given cellular systems.

Although receptor biology is still elusive to a significant extent, a body of research evidence has been gathered around the ligand–receptor interaction and their associated signaling capacities, so that at least three different groups of FFAR1 activating ligand classes emerge: (i) endogenous/orthosteric agonists, e.g., long-chain fatty acids (LCFAs), (ii) partial allosteric agonists (agoPAMs), e.g., fasiglifam/TAK-875, MK-8666 or AM 837, and (iii) full allosteric agonists—e.g., AM 1638, AP8 or compound A [60,61]. These groups differ not only in their apparent binding sites at the receptor, but also in their capabilities to induce different signaling routes downstream of FFAR1, eventually leading to different outcomes on FFA1 receptor phenotypes in vivo. Lin et al. identified at least three binding sites at the FFA1 receptor which partake in allosteric communication with each other as it was shown that members of the above-mentioned ligand groups targeting distinct binding sites provoked allosteric modulation to different extents [62]. The polyunsaturated fatty acid docosahexaenoic acid (DHA), for instance, showed negative allosteric modulation in combination with the partial allosteric agonist AM 837 on the α-factor while this combination showed positive allosteric modulation on the β-factor. However, a composite parameter incorporating both α and β allosteric modulations was shown to be positive for all tested allosteric/orthosteric ligand combinations [62]. A hallmark of FFAR1 biology is its potentiation of glucose-induced insulin secretion on pancreatic β-cells. Fasiglifam, a partial FFAR1 agonist and positive allosteric modulator, couples to Gq with less efficacy compared to full FFAR1 agonists, such as AM 1638 or fatty acids, but is a full agonist for β-arrestin recruitment. The mechanism underlying potentiation of GSIS seems to be different for different agonists. While β-arrestin knock-down significantly attenuated fasiglifam-mediated GSIS and Gq inhibition did not lead to substantial reduction in GSIS, the tested full agonists were sensitive to Gq inhibition but not β-arrestin depletion [58]. 

The ability of the FFA1 receptor to confer GLP-1 secretion from enteroendocrine cells has been another area of intense research with findings suggesting that partial agonists lack the ability to induce GLP-1 secretion, while full agonists or full ago-PAMs showed this additional behavior. Allosteric potentiation of partial agonists by ago-PAMs is necessary to enhance receptor activation and GLP-1 secretion [63]. Hauge et al. proposed that combined signaling via Gs and Gq accounts for this feature as the full allosteric agonists AM 1638 or AM 5262 were more efficacious in GLP-1 induction in mice compared to the partial allosteric agonists fasiglifam or AM 837, which only activate Gq [52]. Whether additional coupling partners of FFAR1 further modulate the biological activity is currently not known, but reports suggest differences between partial and full agonists at the FFA1 receptor also apply to their coupling efficiency to G12 proteins [59]. 

The additional activity on gut hormone release appears to be a promising biological—and potentially also pharmacological—feature that warrants further investigations. Leveraging beneficial incretin effects on food intake and body weight has been shown for FFAR1 full agonists depending on afferent vagal nerve stimulation [64]. The combination of GLP-1 increasing medications, such as DPP4 inhibitors with partial FFAR1 ago-PAMs such as fasiglifam demonstrated improved glycaemic control in diabetic patients compared to placebo or metformin treatment [65]. Synergistic effects of these two targeted therapies were also shown with Astellas’ FFAR1 agonist AS2575959 and the DPP4 inhibitor sitagliptin. The combination of AS2575959 and sitagliptin enhanced GSIS, oral glucose tolerance, and GLP-1 secretion, while the FFAR1 agonist alone could not induce GLP-1 secretion [66]. Researchers from Merck showed that full FFAR1 agonists reduce food intake and body weight in diabetic rodent models, an effect that was dependent on the ability to stimulate GLP-1 secretion. The combination of the FFAR1 full agonist compound A and a DDP4 inhibitor could further improve weight loss in diet-induced obese mice [67]. The same group confirmed a differentiation between partial FFAR1 ago-PAMs and full ago-PAMs by showing that the partial agonist MK-8666 while reducing food intake did not lead to a reduced total body weight in the Goto-Kakizaki rat model of type 2 diabetes, whereas the full ago-PAMs AP1 and AP3 did [68]. 

The most advanced FFA1 receptor agonist in clinical development is the partial ago-PAM fasiglifam, discovered and developed by Takeda [69,70]. Fasiglifam showed promising efficacy in preclinical in vitro and in vivo models and clinical trials with significant reduction in Hba1c while carrying no or low risk of hypoglycemia. However, due to liver toxicity findings, fasiglifam development was stopped before reaching the market [71]. Inhibition of the hepatobiliary transporters MRP2 and OATP as well as NTCP were suggested to be the cause of this finding [72]. After discontinuation of fasiglifam, discussions on the future of FFAR1 as a target for the treatment of metabolic diseases arose [73]. Nevertheless, FFA1 receptor modulators still gain a lot of attention from both academic groups as well as the biopharma industry with new chemical matter available and increased understanding of the target biology that led to refined targeting strategies—e.g., full ago-PAM versus partial ago-PAM or biased agonists. Further medicinal chemistry efforts have led, for example, to improved FFAR1 agonists with less or no liver toxicity findings in preclinical models [74,75]. Recently, researchers from Janssen Pharmaceutica designed an FFAR1 superagonist that showed higher efficacy on the Gs signaling pathway compared with other full agonists, such as AM 1638, and also a higher potency on both the Gs and Gq pathways [76]. While the higher potencies were reflected in lower EC_50_ values in the GSIS assay on human islets and in reduction in food intake and body weight loss in diet-induced obese mice, the impacts and benefits that a superagonist at the Gs pathway can potentially provide will need further investigations. For excellent summaries on medicinal FFAR1 ligand chemistry in the context of clinical or preclinical phase development, the reader is referred to further references [60,61,77]. Despite the setback of a late discontinued FFAR1 allosteric modulator, the current patent landscape and research activities indicate that it is likely that we have not seen the last FFAR1 targeting molecule making its way into the clinic. Table 2 shows a selection of allosteric ligands identified for FFAR1.

## 3. Free Fatty Acid Receptors 2 and 3 (FFAR2 and FFAR3)

### 3.1. Introduction and Molecular Receptor Pharmacology

The FFA2 and FFA3 receptors are activated by short-chain fatty acids (SCFAs), which are the endogenous agonists mainly produced by the gut microbiota via fermentation of indigestible carbohydrates in humans—i.e., dietary fiber. Hence, FFAR2 and FFAR3 overlap in their recognition of orthosteric ligand as both sense SCFAs up to a chain length of six carbon atoms. However, there is a slightly different order of potency for the endogenous agonists. For FFAR2, this is: C2~C3~C4 > C5 > C6~C1 for the human receptor, while C2 has higher affinity compared to C3 for the mouse FFA2 receptor and for FFAR3: C3~C4~C5 > C6 > C2 > C1 [85]. In addition to the naturally occurring SCFAs, synthetic ligands have also been generated mainly with the aim of overcoming the similar ligand recognition pattern to achieve receptor selectivity. Chemical modification led to a group of small carboxylic acids that showed preferential activation of FFAR2 over FFAR3 and thus increased the understanding of orthosteric ligand–receptor interactions of the SCFA sensing FFA receptors [86], which was followed by more selective synthetic orthosteric FFAR2 agonists [87]. A high-throughput screening with subsequent compound optimization identified selective phenylacetamide scaffold-based FFAR2 ligands that were shown to act as allosteric agonists, which both activate the receptor on their own but also enhance the signaling capacities of orthosteric SCFAs at the FFA2 but not the FFA3 receptor [88,89]. Although the exact binding site of the identified prototypical allosteric ligand 4-CMTB could not be unequivocally identified by extensive mutational analysis [90], Smith et al. provided some structural insights into the allosteric mechanism by showing that the extracellular loop 2 (ECL2) of the FFA2 receptor is crucial for conveying allosteric communication from the SCFA binding site to the allosteric binding site of 4-CMTB [91]. Later, Grundmann et al. found that FFA2 receptor activation by 4-CMTB actually functions in a unique manner. 4-CMTB could trigger receptor activation by both the orthosteric as well as the allosteric sites in a temporally staggered fashion, thus characterizing 4-CMTB as the first sequentially activating ligand (SEAL) [92]. It was shown that this process involves interactions with two distinct receptor sites to allow for temporal control of GPCR signaling, a manifestation of what has been denoted as temporal bias [93]. 

FFAR2 selective antagonists that all showed to be inverse agonists by reducing constitutive activities of FFA2 receptors were described in a Euroscreen patent, with CATPB being the best characterized antagonist [94], and later being described as structurally distinct in a high-throughput screening (HTS) campaign including 20,000 compounds (Park et al. [95]). The only FFAR2 antagonist that has reached the clinical development stage is GLPG0974, which was discovered and developed by Galapagos to treat intestinal bowel disease patients via a neutrophil activation reducing mechanism [96,97]. However, further clinical development was discontinued after a lack of efficacy in a phase II clinical study.

While the FFA3 receptor has been reported to only signal via the Gi protein family [20], FFAR2 was found to activate G proteins from both the Gi and the Gq families [20], but induce signaling in a Gαi/o/q/11- and β-arrestin-independent fashion, which was shown to be mediated via Gα12/13 proteins [31]. 

### 3.2. Enteroendocrine Function

SCFAs, such as acetate and propionate, are known to increase GLP-1 secretion in the intestine, which has been proven by Tolhurst et al. to be driven by enteroendocrine L cells in wild-type but not FFAR2 knock-out mice. FFAR3 is also expressed in L cells [98] and is also involved in GLP-1 release; however, FFAR2 likely plays a superior role in this process, suggesting that FFAR3 has a modulatory function [99,100]. In a valuable study making use of a designer receptor that is exclusively activated by designer drugs (DREADD) approach, a DREADD-FFAR2 antagonist was sufficient to block GLP-1 release in a DREADD-FFAR2 knock-in mouse model [101]. Gq protein signaling was noted to be mainly responsible for conveying the GLP-1 releasing effect by FFAR2 while inhibition of Gi protein signaling by use of pertussis toxin (PTX) only moderately and nonsignificantly reduced GLP-1 release from primary colonic cultures. Gi signaling, however, is thought to negatively modulate incretin secretion with somatostatin blocking cAMP-mediated GLP-1 secretion. However, knock-out of Gi protein-coupled FFAR3 in colonic cultures showed a blunted GLP-1 release, which could not be further rationalized [99]. More recently, an entanglement of Gq and Gi protein signaling mechanisms was described for GLP-1 release from colonic crypts. Propionate induced a Gi/p38 signaling route deriving from early endosomes after internalization in a β-arrestin-dependent way. This pathway, however, also involved Gq proteins as the selective Gq protein inhibitor YM-254890 partially inhibited GLP-1 release. In fact, Gq protein inhibition partially inhibited cAMP decreases mediated by propionate in an FFAR2-dependent but FFAR3-independent fashion, providing evidence of a role of FFAR2 as a spatiotemporal controller of a concerted Gi/q signaling network [100]. The involvement of Gq proteins in the control of the Gαi/o-βγ-PLC pathway toward intracellular calcium rise has recently been described by the Kostenis group [102].

### 3.3. Pancreatic β-Cells

FFAR2 and FFAR3 have been found to reside on pancreatic β-cells and regulate insulin secretion. While it is known that Gq protein signaling promotes and Gi protein signaling decreases GSIS of β-cells, it is not surprising that the Gq/i protein-coupled FFA2 receptor both positively and negatively regulates GSIS and that the Gi protein-coupled FFA3 receptor is generally linked to negative regulation of insulin secretion [103,104,105,106,107,108]. This divergence in G protein signaling downstream of FFAR2 has led to hypotheses on potential biased ligands that evoke distinct pharmacological profiles. In fact, small carboxylic acids recognized to activate FFAR2 orthosterically promote GSIS via a PLC inhibitor-sensitive Gαq/11-mediated pathway, whereas allosteric activation with 4-CMTB [88] inhibited GSIS in a PTX-sensitive and Gαi/o-dependent fashion [107]. Another allosteric activator of FFAR2 (compound 58 [89]), although active on both Gq and Gi protein pathways, showed a Gαq/11-dependent potentiation of GSIS in mouse and human islets [105]. These results indicate that balancing inhibitory and stimulatory signaling pathways downstream of FFAR2 or even fine-tuning of G protein interaction with the cellular signaling machinery upstream of the second messengers, IP_3_/Ca^2+^ and cAMP, might prove useful in shaping insulinotropic function. Interestingly, allosterically activated FFA2 receptors did not desensitize after prolonged stimulation and insulin secretion was persevered even after sustained allosteric stimulation [105]. Whether biased signaling downstream of FFAR2 can be exploited for therapeutic applications by a targeted approach, e.g., rationally designed biased ligands, will have to be assessed in future studies. 

### 3.4. Metabolic Functions 

In adipose tissue, only the FFA2 receptor is expressed and was shown to inhibit lipolysis in a PTX-sensitive manner, demonstrating that Gi protein signaling is the key driver of this event [101,109]. The involvement of FFAR2 in the regulation of metabolic homeostasis was shown at the level of adipocytes and insulin signaling of the receptor. Several reports indicate that FFA2 receptor deficiency entails obesity, while FFAR2 overexpression in adipose tissue protects mice from obesity even under high-fat diets [35,110]. FFAR2-mediated signaling suppressed insulin signaling to inhibit adipogenesis as well as promoted metabolism of lipids and glucose in muscle tissue [111]. Another study found a link between ketone bodies and FFA2 receptor-mediated energy hemostasis and lipid metabolism. Systemic levels of ketone bodies increase following ketogenic diets or after starvation as a physiological reaction to counteract loss of energy sources via, e.g., carbohydrates. Intermittent fasting is a well exploited weight-loss approach that has recently gained popularity. In addition to the major ketone body β-hydroxybutyrate, acetoacetate also increases after starvation or ketogenic diet. Miyamoto et al. found that acetoacetate is a full agonist at the FFA2 receptor and mediates weight loss and lipoprotein lipase activity under nutritional shifts. Thus, FFAR2 wild-type, but not knock-out, mice showed substantial weight loss after intermittent fasting, suggesting a central axis of the ketone body acetoacetate and FFAR2 in metabolic regulation [112].

Improved metabolic body homeostasis in the forms of glucose tolerance and enhanced fatty acid oxidation was also observed in wild-type, but not FFAR2 knock-out, animals fed a high-fat diet, which received a probiotic bacterial strain that increased intestinal SCFA levels [113]. Further evidence for a protective role of FFAR2 signaling in the context of energy metabolism comes from a study that showed beneficial effects of fermentable carbohydrates on food intake and obesity. A diet supplemented with inulin led to an increase in peptide YY (PYY) producing cells and finally ameliorated body weight control and glucose homeostasis via an FFAR2-dependent mechanism [114]. However, this view is contradicted by an earlier study demonstrating that FFAR2 knock-out mice were protected from high-fat diet (HFD)-induced metabolic aberrations [115]. Another study showed that protective effects of the dietary fiber inulin on HFD-induced obesity is preserved in FFAR2-deficient mice or mice receiving β-acids that decrease bacterial-derived SCFAs, indicating an FFAR2- and SCFA-independent mechanism of protection. The authors propose an IL-22 signaling pathway from innate lymphoid cells (ILC3) to drive enterocyte proliferation, mucosal defense mechanisms to combat microbiota encroachment, and colon atrophy that finally leads to low grad inflammation and metabolic syndrome [116]. On the other hand, key functions of ILC3s for the maintenance of gut homeostasis and host defense against pathogens was shown to be dependent on the FFA2 receptor as FFAR2 deficiency decreased ILC3-derived IL-22 and led to increased susceptibility to bacterial attack by *C. rodentium* and colitis induction. The critical IL-22 increase reported here was mediated by an FFAR2-Akt/MAPK/STAT3/RORγt signaling pathway in ILC3s [117]. 

### 3.5. Inflammation and Immune Function

Inflammation is a shared (patho)physiological mechanism in the amalgamative disease fields of metabolic dysregulation, infectious diseases, and (auto-)immune diseases, such as intestinal disorders. FFA2 receptor activation on dendritic cells by SCFAs derived from the gut microbiota triggers antibody production by B cells by increasing BAFF and ADLH1a2 expressions, which leads to elevated retinoic acid levels [118,119]. These mechanisms that underlie the beneficial effects on intestinal homeostasis relied on FFAR2 expression as knock-out animals did not response in a similar fashion and were reported to be independent of T cells [119]. Another study placed focus on T regulatory (T_reg_) cells in the microbial SCFA-mediated immunomodulatory effects of FFAR2. SCFAs regulated the abundance and activity of this cellular population to finally protect mice against colitis induction in the T cell-transfer model [120]. Furthermore, T_reg_ cells have be found to be immunoregulatory during kidney allograft transplantation. Wu and colleagues could show that mice receiving a kidney allograft, but not an isograft, exhibited gut microbial dysbiosis and allograft rejection when fed a normal chow diet, which was prevented by feeding a high-fiber diet or a normal chow diet supplemented with acetate. These beneficial effects were mediated by T_reg_ cells in an FFAR2-dependent manner, as FFAR2-deficient mice were not protected [121]. In addition to hematopoietic cells, renal resident cell types such as tubular epithelial cells and podocytes have also been found to express FFAR2 and react with reduced proinflammatory and profibrotic gene expressions upon hyperglycemic insult when FFA2 receptors were stimulated. Antifibrotic, anti-inflammatory, and kidney function preserving effects of SCFAs such as acetate and butyrate or high-fiber diet via changes in the gut microbiome have been demonstrated to be mediated by FFA2 and GPR109A receptors for mice suffering from streptozotocin (STZ)-induced diabetic nephropathy [122]. Renal mesangial cells have also been shown to express FFAR2 and stimulation with orthosteric and allosteric ligands for FFAR2 led to reduced oxidative stress and proinflammatory gene transcription by downregulation of NFκB signaling in an FFA2 receptor -dependent manner. This process was suggested to underlie the protective effects of SCFA-mediated amelioration of renal dysfunction in STZ-induced diabetic nephropathy in mice [123]. Additionally, the FFAR2 specific allosteric agonist CFMB had decreasing effects on TNF-α-induced MCP-1 upregulation in cortical epithelial cells, indicating an anti-inflammatory effect of FFA2 and FFA3 receptor signaling in renal diseases [124]. Protection from westernized diet-induced cardiovascular disease by SCFA supplementation or resistant starch diet to wild-type mice but not mice deficient of the SCFA receptors FFAR2, FFAR3, and GPR109A was recently reported. Kaye et al. assessed cardiovascular endpoints such as cardiac function, hypertrophy and fibrosis, renal sodium and potassium excretion, and blood pressure and provided evidence for a prominent role of FFAR2 and GPR109A in mediating a T_reg_ cell driven immune-modulatory effect to protect from detrimental effects of low-fiber diets on cardiovascular health [125]. 

Fachi et al. documented the involvement of ILC3s in the balancing act of FFA2 receptor signaling in pro- and anti-inflammatory reaction by showing how the short-chain fatty acid acetate mediates IL-22 secretion by ILC3s in an FFA2 receptor-dependent and IL-1 receptor involving way. FFA2 receptor-enhanced neutrophil sensitization and secretion of IL-1β together with the immune-modulatory function via ILC3s were shown to underlie the protective value of acetate by defending a *Clostridium difficile* infection [126]. A central role of the SCFA sensing receptor FFAR2 in the tight link between the gut microbiome and host health state has thus been repeatedly confirmed. 

Protection of host integrity from pathogen infections has also been shown for the airway system. *Klebsiella pneumoniae* lung infection was hindered by increased phagocytotic activity of neutrophil and FFAR2 activity of macrophages [127]. Sencio et al. investigated the role of gut microbiota in the controls of *Streptococcus pneumoniae* superinfection postinfluenza and found FFAR2 function on alveolar macrophages to be critical in reducing susceptibility of infected mice to bacterial respiratory superinfection [128]. 

Internalization of stimulated FFA2 receptors affects cellular entry and following replication of the influenza A virus. siRNA knock-down of FFAR2 or of internalization machinery components such as β-arrestin 1 or the β2-subunit of the AP-2 complex reduced receptor internalization and thus virus endocytosis [129]. While regulation of FFA2 receptor surface abundance is an elegant mechanism to confer cellular sensitivity, it will require further investigation to elucidate whether this mechanism functions as a means to safeguard the host system from pathogen attacks under different circumstances.

Polymorphonuclear cells, especially neutrophils, show notably high expression levels of FFA2 receptors, which indicates a prime role of this receptor in neutrophil behavior. Indeed, several reports clearly demonstrated that FFAR2 promotes chemotaxis in mouse and human neutrophils. Beyond migratory response, FFAR2 also accounts for other central neutrophils traits, such as adhesion, rolling, transmigration, phagocytosis, and killing mechanisms, such as granule formation and trafficking or oxidative burst [20,130,131,132,133,134,135]. In addition to an offensive first line host defense, neutrophils also regulate and end overt inflammatory host responses. An imbalance in the pro- and anti-inflammatory regulation of neutrophils underlies various inflammatory diseases, such as intestinal bowel disease, asthma, cardiovascular diseases, arthritis, psoriasis or sepsis. The exact mechanisms of FFAR2 in balancing this bipotential activity, however, are still largely elusive. Indeed, FFA2 receptor signaling has been linked to both pro- and anti-inflammatory neutrophil phenotypes. Sina et al. found that FFAR2 knock-out mice responded with reduced neutrophil recruitment, diminished proinflammatory cytokine production, and less tissue degradation in a chronic setting of dextran sulfate sodium (DSS)-induced colitis [132]. Phagocytic cells from mice deficient in FFAR2 also demonstrated lower inflammasome activation and reduced proinflammatory cytokine production in vivo in a monosodium urate crystal-induced gout model [136]. A systemic inflammation model by lipopolysaccharide (LPS) challenge revealed that a transient change in neutrophil activation and recruitment from a state of acutely decreased rolling and adhesion one hour after LPS administration to a state of increased neutrophil recruitment and migratory phenotype toward fMLP (N-Formylmethionine-leucyl-phenylalanine) four hours after LPS challenge in FFAR2-deficient mice. Herein, FFAR2 activation led to reduced migration in wild-type but not in FFAR2 knock-out mice, indicating an immune regulatory function of FFAR2 [137]. FFAR2 has also been implicated in regulatory macrophage activation, as M2- but not M1-type macrophages from adipose tissue were activated by FFAR2, presumably accounting for the tissue remodeling and anti-adipogenic effect of M2 macrophages in chronic inflammatory and insulin desensitization diseases such as obesity [138]. The role of FFAR2 in the resolution of inflammation and late stage outcomes in different disease models, ranging from arthritis over colitis to asthma, has been demonstrated by Maslowski et al. [131]. FFAR2-deficient mice showed exacerbated inflammation in all models concomitant with increased proinflammatory cytokine production and increased peritoneal neutrophil infiltration after *Streptococcus aureus* challenge and increased mortality in the chronic colitis model, suggesting a protective role of FFAR2 in the management of overt inflammatory processes [131]. Interestingly, a drastic increase in mortality in an acute colitis model by DSS challenge over six days in the study of Sina et al. also points to a beneficial immune regulatory function of FFAR2 [132]. The disparate results for FFAR2 in the chronic colitis model may stem from differences in the respective experimental setups involving different timings for the sequences of DSS administration as well as different concentrations of DSS (reviewed in [139]). In general, it becomes clear that a balanced response of neutrophils is especially important in the management of exuberant inflammation as it occurs in acute and chronic disease states. 

Since neutrophil responsiveness is fundamental for all of the above-mentioned immune-modulatory functions of this cell type, it is not surprising that several studies investigated the sensitivity and capability of FFAR2 signaling to evoke a neutrophil response. It was found that TNFα can prime neutrophils and increase subsequent FFAR2 orthosteric agonist responses [140]. Allosteric targeting of the FFA2 receptor on neutrophils could enhance orthosteric FFAR2 activation and boost the NADPH oxidase response [135], but could also allosterically increase ATP-induced P2Y_2_ receptor-mediated neutrophil activation using a Gαq/11-dependent signaling pathway downstream of FFAR2 [134]. Occupancy of the neutrophil FFA2 receptor by two allosteric ligands that do not activate the receptor when in isolation produced an NADPH response when combined. Interestingly, orthosteric FFA2 activation by SCFA propionate or heterologous receptor activation by ATP, for instance, still led to calcium rise and concomitant NADPH oxidase activation [141]. These experiments show that allosteric and orthosteric FFA2 stimulation are interdependent but induce distinct downstream signaling events. Moreover, Frei et al. revealed distinct signaling compositions of Gαi/o- and Gαq/11-dependent as well as Gi/q-independent pathways between orthosteric and allosteric FFAR2 activations in neutrophils with propionate and 4-CMTB, respectively. Furthermore, it was shown that allosteric targeting of FFAR2 can resensitize the receptor and render previously silenced neutrophils selectively responsive toward orthosteric FFAR2 activation [133]. Although the molecular mechanisms underlying the allosteric modulation of FFAR2 still need to be fully elucidated, the phenomenon of allosteric rescue of a desensitized FFA2 receptor could potentially open novel avenues of drug discovery approaches.

### 3.6. Nervous System

The FFA3 receptor is expressed in the peripheral nervous system in submucosal and myenteric ganglia of the small intestine and also proximal colon as well as in sympathetic ganglia [98,142]. FFAR3 activity on myenteric neurons has been linked to decreased gut motility and antisecretory effects after cholinergic or serotonergic stimulation by usage of the FFAR3 synthetic agonist AR420626 [143]. On sympathetic neurons, FFAR3 activation is believed to increase energy expenditure and heart rate by crosstalk with the adrenergic receptor system via Gi/o proteins since isoproterenol or PTX pretreatment abolished the effects as well as antagonism of FFA3 receptors by β-hydroxybutyrate [142]. A role of FFAR3 in the gut-cardiovascular-energy homeostasis axis was corroborated in a further study, demonstrating that offspring of FFAR3-deficient mice had lower body temperatures and heart rates [144]. 

Table 3 summarizes FFAR2 and FFAR3 allosteric ligands described in the literature.

## 4. Free Fatty Acid Receptor 4 (FFAR4)

FFAR4, formerly called GPR120, is next to FFAR1, the second receptor activated by long-chain fatty acids. α-Linolenic acid (18:3(n-3)) (aLa) and linoleic acid (18:2(n-6)) activate FFAR4 with EC_50_ values of 0.4 to 2 and 5 µM, respectively. However, there are further free fatty acids similar to stearidonic acid, eicosapentaenoic acid (EPA), and docosahexaenoic acid (DHA) known to activate FFAR4 with an EC_50_ below 5 µM [148]. Together with aLa, the latter are members of the ω-3 family of fatty acids that have been associated with anti-inflammatory effects [149,150]. 

Agonism at FFAR4 triggers two different signaling pathways. The receptor is known to couple, via Gαq/11 proteins, to the production of inositol trisphosphate (IP_3_) and Ca^2+^ release [151,152]. This particular signaling of the receptor has been linked, for instance, to its function in GLP-1 release from enteroendocrine cells in the colon [23]; the extent of the contribution by FFAR4 to GLP-1 release has been questioned [63,153]. FFAR4 also couples to Gαi/o and the inhibition of adenylate cyclase activity was shown by Engelstoft et al. [28] in the example of ghrelin secretion inhibition for primary gastric mucosal cells. Another mechanism employed by FFAR4 is based on binding to β-arrestin 2. The FFAR4/β-arrestin 2 complex binds transforming growth factor-β-activated kinase 1 (TAK1) binding protein 1 (TAB1) and thereby sequesters it from TAK1. This subsequently leads to a reduction in the TAB1/TAK1 complex and its proinflammatory activity at the conversion point of the toll-like receptor (TLR) and tumor necrosis factor (TNF) α receptor signaling pathways [154] in macrophages. 

Human FFAR4 is expressed as a short splice variant of 361 aa and a long isoform carrying an additional 16 aa in the third intracellular loop (ICL3). For the latter, it has been shown that it is, compared to the short isoform, unable to mediate agonist-stimulated Ca^2+^ release, whereas β-arrestin binding and downstream internalization as well as trafficking are comparable for the two splice variants. Because the long isoform is not found in other species such as rodents and even cynomolgus monkeys [151,155], the physiological consequence of this species difference is so far unknown. 

FFAR4 expression is detected in the colon, in enteroendocrine cells, adipocytes, and macrophages. In addition, a high expression level was also found in the lung [23]. Here, expression is localized to the airway epithelium and experiments with FFAR4 knock-out mice suggest that the receptor promotes epithelial repair after lung injury [156]. The pancreas is a further organ where FFAR4 is expressed and it has been shown to be protective against palmitate-induced apoptosis. In adipocytes, FFAR4 stimulation enhances GLUT4 translocation and glucose uptake [154]; moreover, this receptor has been suggested to be involved in the differentiation and development of adipocytes [157,158]. Houthuijzen et al. [159] have shown that FFAR4 is expressed in splenic macrophages and is related to the development of chemotherapy resistance by regulating the production of the resistance inducing lipid mediator lysophosphatidylcholine (24:1). For murine RAW 264.7 cells and also for intraperitoneal macrophages, anti-inflammatory effects of FFAR4 activation point toward an interesting mechanism in the context of inflammation and insulin resistance, which has been closely examined by different groups [154,160,161]. Oh et al. [154] demonstrated an inhibition of the proinflammatory activities of TLR-2, -3 and -4 agonists as well as for TNFα for ω-3 fatty acid DHA and RAW 264.7 cells. This effect was abrogated by FFAR4 and β-arrestin 2 siRNA-mediated knock-down, whereas Gαq/11 and β-arrestin 1 reduction caused no effect. Co-immunoprecipitation experiments supported the above-described FFA4 receptor agonist-dependent association of β-arrestin 2 with TAB1 and the entailing reduced amount of proinflammatory TAB1/TAK1 complexes. The broad anti-inflammatory effects of FFAR4 agonism could also be seen for LPS-stimulated macrophages with phosphorylation of IKK, JNK, and MCP-1 as well as IL-6 secretion being dependent on the presence of the FFA4 receptor. Anti-inflammatory FFAR4 effects were also seen in 3T3-L1 adipocytes. For these cells, as well as in primary adipose tissue cultures, DHA-dependent translocation of GLUT4 to the plasma membrane and enhanced 2-deoxyglucose uptake could be demonstrated. In this case, FFAR4 and Gαq/11 knock-down abolished enhanced translocation and uptake, whereas the reduction in β-arrestin 1 and β-arrestin 2 levels by siRNA showed no effect [154]. The latter authors supported their results by confirming them with the FFAR1/FFAR4 agonistic tool compound GW9508, which is more of an FFAR1 agonist than an FFAR4 agonist [162], necessitating the reliance on negative expression results for FFAR1 in macrophages and adipocytes. 

Subsequently, Shimpukade et al. [163] reported, using TUG-891, a more selective human FFAR4 agonist as a tool compound, for which, based on studies mutating relevant amino acids in the orthosteric binding pocket including Arginine at position 99 (Arg99), it could be shown that this compound is an orthosteric agonist [164]. This improved FFAR4 agonist was characterized in detail by Hudson et al. [160]. In a recombinant cell line, this compound showed an EC_50_ of 60 nM at hFFAR4 and 17 µM EC_50_ at hFFAR1 in a β-arrestin 2 recruitment assay, displaying a 283-fold selectivity. This was confirmed in principle in a Ca^2+^ release assay format (95 nM EC_50_ hFFAR4, 5 µM EC_50_ hFFAR1); in these recombinant assays, TUG-891 showed the same efficacy as the endogenous agonist aLa. The potency of TUG-891, as well as that of the endogenous agonist aLa, was much lower at the endogenous FFAR4 of the colon cancer cell line HT29 with an EC_50_ of 4.4 µM for TUG-891 and 33 µM for aLa in the Ca^2+^ release assay with different levels of receptor reserve being a possible underlying cause for this pronounced difference. For the mouse orthologous receptors, there was a 61-fold selectively regarding the β-arrestin 2 recruitment assay format (19 nM EC_50_ mFFAR4, 1.2 µM EC_50_ mFFAR1); however, there was almost no selectivity for the two orthologous receptors in the Ca^2+^ release assay (128 nM EC_50_ mFFAR4, 389 nM EC_50_ mFFAR1). Based on TUG-891 as a tool compound, Hudson et al. [160] re-examined the different pharmacological effects ascribed to FFAR4 when stimulated with LCFAs. For two mouse enteroendocrine cell lines, STC-1 and GLUTag expressing both FFAR1 and FFAR4, TUG-891 addition caused an increase in GLP-1 secretion similar to aLa, while a selective FFA1 receptor agonist only marginally increased GLP-1 secretion and an FFA1 receptor antagonist only slightly decreased the amount of GLP-1 induced by aLa. This supports the view that FFAR4 is the main FFA receptor responsible for stimulation of GLP-1 secretion with only minor contributions by FFAR1, at least for these two cell lines. Two further effects ascribed to the agonism of FFAR4 were examined with the tool agonist, whereby it could be demonstrated that TUG-891 and aLa both stimulated the increase in glucose uptake by adipocytes; however, this has a low potency when compared to the EC_50_ values derived from the recombinant systems, but is comparable to the results based on the endogenously expressed receptor in HT29 cells. The efficacies at the highest agonist concentrations employed were modest compared to 1 µM insulin (557%). aLa reached 92% and TUG-891 47% increase in 2-deoxyglucose uptake; however, higher agonist concentrations might have been slightly more efficacious, as suggested by the dose response curves. In addition, the broad anti-inflammatory effect of FFAR4 agonism was examined with the specific synthetic FFAR4 agonist TUG-891 in the example of LPS-induced TNFα secretion by the monocytic mouse cell line RAW264.7. Both TUG-891 (10 µM) and aLa (100 µM) inhibited TNFα secretion by 30% and 29%, respectively; however, this was also at a low potency, comparable with the results obtained with HT-29 cells or adipocytes and endogenously expressed receptors. The ω-3 fatty acid DHA (100 µM) reached a higher efficacy of 88% TNFα secretion inhibition; however, DHA might exert additional anti-inflammatory effects besides FFAR4 agonism via cyclooxygenase 2 or peroxisome proliferator-activated receptor (PPAR)γ [165,166]. With the use of TUG-891, Hudson et al. [160] confirmed the anti-inflammatory and insulin-sensitizing effects of FFAR4 agonism; on the other hand, the limited selectivity of the compound between murine FFA1 and FFA4 receptors and its rather low potency when looking at endogenously expressed FFA4 receptors are two obstacles that prevent its use in translating in vitro results into in vivo effects. 

A further substance with high human free fatty acid receptor 4 selectivity and which could also be used for in vivo models is CpdA [161,167]. This compound is a very potent agonist at human FFA4 receptors in a Ca^2+^ release assay (EC_50_ 24 nM) with no significant activity on the human FFA1 receptor up to 1 µM. In an IP_3_ production assay, it has a very similar potency at the human and mouse FFA4 receptors; the same is true for a β-arrestin recruitment assay where CpdA stimulates both human and mouse orthologues with an EC_50_ of 350 nM. There are no data shown for the murine FFA1 receptor or cells with endogenous expressions of FFAR4; therefore, the selectivity of the compound for the mouse receptors or its activity for relevant endogenous receptor expression cannot be assessed. In vitro, CpdA (10 µM) inhibited LPS-induced NFκB activation in wild-type mouse macrophages by 50%, while DHA (100 µM), in comparison, had a stronger effect of 75% on reduction in NFκB activation with both effects being dependent on the presence of FFAR4. To examine the in vivo effects of CpdA, mice were fed with high-fat diet (HFD) plus and minus a supplementation with 30 mg/kg CpdA; the treatment regimen was applied in parallel to wild-type and FFA4 receptor knock-out mice. For the wild-type, but not the knock-out animals, CpdA significantly improved glucose and insulin tolerance as well as decreased insulin secretion. An increased insulin sensitivity after CpdA treatment in wild-type mice could also be shown after insulin injection by increased Akt phosphorylation in muscle and liver tissues. The authors could also show that CpdA, although somewhat less effective than DHA, reduced macrophage infiltration into adipose tissue. As a marker of systemic inflammation, levels of different circulating cytokines were determined, and CpdA selectively and significantly reduced these levels in wild-type animals. However, CpdA had no measurable effects on the secretion of GLP-1 nor in isolated Langerhans islets on glucose-stimulated insulin secretion. The effects of CpdA on the 2-deoxyglucose uptake of adipocytes were rather modest, and the increase elucidated by DHA was even smaller [161,167]. In summary, the experiments performed with CpdA prove the anti-inflammatory and insulin-sensitizing properties of FFAR4 agonists. 

Most of the compounds described as FFAR4 agonists contain a carboxylate or bioisostere, which have been suggested to interact with Arg99 of the receptor in the orthosteric binding pocket. With GSK137647 [168], a sulfonamide-based and the first nonacidic FFAR4 agonist was described. The authors provide solid evidence, based on mutational studies, that the compound nevertheless binds in the orthosteric binding pocket. In this context, it is of interest that, so far, the only available FFAR4 antagonist, AH-7614, first described as compound 39 [169], with an IC_50_ of 20 nM from the same compound class, was described as a negative allosteric modulator (NAM) by Watterson et al. [30]. The classification as a NAM is based on the dose-dependent efficacy reduction of four different agonists, TUG-891 and CpdA as acidic FFAR4 agonists and of TUG-1197 and GSK137647 as sulfonamide-based agonists, rather than shifting the potencies of these agonists. In addition, the antagonistic effect was saturable, leaving residual agonistic activity even at the highest AH-7614 concentration, with the exception of GSK137647A, whose agonism could be completely blocked. Apparently, the introduction of a xanthene group into the sulfonamide chemotype induced high-affinity binding of the compound at a site different from the orthosteric sulfonamide agonist site. So far, the antagonistic AH-7614 is the only allosteric ligand described for FFAR4. 

The development of binding assays might be helpful to further examine binding sites and characteristics of FFAR4 ligands, with respect to variable potency and efficacy results depending on the assay employed. In conclusion, the stimulation of FFAR4 with compounds from different structural classes seems to induce strong anti-inflammatory and thereby insulin-sensitizing effects [161,170]. This is supported by genetic data from both human and mice. A R270H mutation strongly reduced FFA4 receptor signaling in a Ca^2+^ release assay after aLa stimulation and this variant was also associated with obesity in a genotyping study of obese and control individuals. In addition, it could be shown by the same authors that FFA4 receptor knock-out animals fed with a high-fat diet developed obesity, glucose intolerance, and fatty liver [171]. 

In summary, these findings stimulated the search for selective and potent FFA4 receptor ligands by academic groups and the pharmaceutical industry, documented by peer-reviewed journals and numerous patents, which have been comprehensively reviewed in recent years [172,173].

In the course of developing novel FFAR4 agonists, TUG-891 was discovered as a derivative of FFAR1 agonists [163]. The ortho-biphenyl core of TUG-891 was then the starting point for further compound optimization programs yielding more selective and, for in vivo studies, suitable FFAR4 agonists [174,175]. Further, FFAR4 agonists were derived from PPARγ agonists [176] or discovered in HTS campaigns and subsequently optimized [177,178]. For several of these variant chemotypes, compounds with favorable pharmacokinetic properties could be developed, enabling in vivo experiments to study their effects as antidiabetic or weight-lowering drugs. These studies are reported in the patent literature or peer-reviewed journals confirming the insulin-sensitizing [175,177,178] or weight-reducing [175] effects of FFAR4 agonists in rodents. In spite of these encouraging data from relevant in vivo models, so far no clinical trials have been reported for FFAR4 agonists; therefore, the development of an FFAR4 agonist suitable for treatment of patients is still awaited.

Table 4 shows the FFAR4 allosteric ligand described in the literature.

## 5. GPR84

GPR84 is a GPCR activated by medium-chain fatty acids (MCFAs), coupling via Gαi/o to the inhibition of adenylate cyclases. As shown by Wang et al. [26], saturated free fatty acids with chain lengths between 9 and 14 carbons activate the receptor, the most potent being capric acid at an EC_50_ of 4.5 µM. In a radioligand binding assay, Köse et al. showed that this MCFA binds with a Ki of 1.78 µM to GPR84. In addition, similar to other receptors that respond to either short-chain or long-chain fatty acids, GPR84 carries an arginine in the putative MCFA binding site at position 172 of the amino acid sequence [179]. Mutating this residue to alanine abolishes the agonism of MCFAs, but not of 3,3′-diindolylmethane (DIM), an agonist acting at an allosteric site of GPR84 [180,181]. So far, however, there is no clear physiologically meaningful connection between GPR84 and MCFAs [182], which would support their role as endogenous orthosteric ligands. In addition, there are three separate ligand binding sites for this known receptor, two for agonists, and one for which only antagonist binding has been described [180]. Therefore, it might be that there are further endogenous or exogenous agonists for GPR84, which are even more relevant for its physiological function than MCFAs. As a consequence, it is currently not proven which of these sites are allosteric or orthosteric.

In addition to the classical MCFA agonist caprylic acid, several other agonists of the MCFA binding site have been described: embelin (2,5-dihydroxy-3-undecyl-1,4 benzoquinone), 6-OAU (6-(octylamino)pyrimidine-2,4(1H,3H)-dione, and alkylpyrimidine-4,6-diol derivatives have been allocated to this binding pocket. Embelin activates GPR84 with an EC_50_ of 200 nM [180]; however, it is a nonselective compound, also acting, for instance, as a potent antagonist of the CXCR2 and adenosine A3 receptor [183]. 6-OAU activates the recombinant receptor with an EC_50_ of 105 nM and elucidates a proinflammatory response of macrophages, which is abolished for macrophages from GPR84 knock-out animals or in the presence of a GPR84 antagonist [184,185]. Among the derivatives of 6-OAU is a very potent agonist, PSB-1584 (Pillaiyar, T. et al., 2018), which in its tritiated form binds GPR84 with a Kd of 2 nM [186]. A very potent GPR84 agonist within the class of alkylpyrimidine-4,6-dioles was discovered in an HTS campaign [187], with ZQ-16 showing an EC_50_ of 213 nM in a recombinant GPR84 assay. The GPR84 agonist DIM [181] retained activity at the R172A mutant receptor and was shown to act as a positive allosteric modulator for agonists binding at the MCFA binding site [180]. DIM is a metabolite of indole-3-carbinol present in several vegetables; however, based on its rather high clearance [188], it is unlikely to play a role as a GPR84 agonist or PAM under normal conditions. Therefore, the physiological agonist or PAM at the DIM binding site of GPR84 still needs to be discovered. 

Antagonists of GPR84 of the dihydropyrimidinoisoquinolinone class [189] are noncompetitive inhibitors of compounds acting agonistically at both the MCFA and DIM binding sites of GPR84 [180]. This finding defines a third ligand binding site on GPR84, for which so far no agonistic or positive allosteric modulatory compounds are known. 

GPR84 expression is highest in granulocytes [25]; however, it is also found on innate immune cells such as macrophages [184]. Low-level expression was detected in liver tissue [186], adipocytes [190], bronchial epithelial cells [191], heart and skeletal muscles [192] as well as microglia [193]. Common to all tissues and cell lines showing GPR84 expression is a strong up to 100-fold upregulation of the receptor by inflammatory stimuli such as LPS for myeloid immune cells and hepatocytes [185], but also systemically in mice where LPS injection was shown to upregulate GPR84 in adipose tissue, bone marrow, brain, lung, kidney, and intestine [184]. Similar upregulation of GPR84 was found by these authors in mice under conditions of hyperglycemia and hypercholesterolemia. At the cellular level, macrophages from the M1-like proinflammatory state were found to express high levels of GPR84 [184]. Furthermore, in human and mouse nonalcoholic fatty liver disease, GPR84 was upregulated as well as in models of kidney injury, such as 5/6-nephrectomy or doxorubicin-induced nephropathy [33,194]. These authors could show that activation of GPR84 on neutrophils by MCFAs as well as embelin leads to chemotaxis. For LPS-primed macrophages, stimulation of GPR84 by 6-OAU was linked to increased levels of inflammatory mediators such as TNFα, IL-6 and CCL2 [184]. In summary, there is a clear link between GPR84 and an enhanced inflammatory host response, making the receptor an attractive target for anti-inflammatory therapies. A first trial for the treatment of ulcerative colitis with the GPR84 antagonist GLPG1205 did not reach sufficient efficacy [195]. However, this compound showed positive results in the phase II PINTA clinical trial for the treatment of idiopathic pulmonary fibrosis (IPF) (NCT03725852). For PBI-4050, a combined GPR84 antagonist and FFAR1 agonist, another FFA receptor targeting compound, was investigated for the treatment of IPF, with positive data acquired from an exploratory phase II study [196]. PBI-4050 treatment in preclinical models of diabetic nephropathy has also been shown to prevent kidneys from deteriorating function and fibrosis [197]. Another study could further show that PBI-4050 reduced renal damage in wild-type but not FFAR1 knock-out mice suffering from adenine-induced kidney injury, suggesting that therapeutic efficacy of this compound relies on the activation of FFAR1 rather than inhibition of GPR84 in this model [198]. 

Table 5 shows the GPR84 allosteric ligands described in the literature.

## 6. Structural Considerations

With the reported involvement of FFARs in energy and metabolic homeostasis, they are interesting targets for pharmaceutical intervention. For designing potent and selective modulators of FFARs, elucidation and understanding of the structural foundation of ligand binding in FFARs is of major importance. 

Multiple allosteric binding sites of FFAR1 have been suggested by radioligand binding studies before crystallographic structural elucidation of the receptor–ligand complexes became available. A network of allosteric interactions between the different sites has been documented—e.g., positive co-operativity between partial and full allosteric agonists [62,199,200]. Positive co-operativity and synergistic effects have also been documented for the partial allosteric agonist TAK-875 and the endogenous orthosteric full agonist γ-linoleic acid in a Ca^2+^ flux assay [70]. 

So far, four crystallographic structures of FFAR1 have been published. Structural elucidation of the remaining FFA receptors has not yet been carried out. In all published structures, FFAR1 adopted the fundamental structure of GPCRs consisting of a seven-transmembrane helices bundle. Srivastava et al. and Lu et al. reported two complexes employing the partial allosteric agonists TAK-875 and MK-8666, respectively [81,201]. Lu et al. furthermore provided the structure of a ternary complex consisting of a receptor, partial allosteric agonist MK-8666, and full allosteric agonist AP8, whereas Ho et al. published a binary complex of FFAR1 together with the full allosteric agonist compound 1 [81,84].

All crystallographic structures are based on stabilized human FFAR1 constructs. The most commonly applied changes to the sequence are a T4 lysozyme protein inserted into the third intracellular loop accompanied by four point mutations in the transmembrane array of helices (Leu-42^2.40^Ala, Phe-88^3.34^Ala, Gly-103^3.49^Ala, and Tyr-202^5.58^Phe) [201,202]. Several observations regarding the behavior of this mutant FFA1 receptor have been made. From comparison with the helix architecture of other known GPCR structures, Srivastava et al. deduced that these mutations seem to constrain the receptor in an inactive conformation [201]. Ho et al. found reduced binding of [^3^H]-AM-1638 to the stabilized receptor when compared to the wild type FFAR1 [84] and Lu et al. documented a five-fold decrease in the dissociation constant Kd of the partial allosteric agonist MK-8666 when bound to the thermostabilized mutant receptor, while co-operativity with the full allosteric agonist AP8 was maintained [81]. Ternary complex crystals of the FFA1 receptor with the two allosteric ligands MK-8666 and AP8 were not obtained, with mutant receptors carrying the four point mutations. Instead, a similar construct without the Phe-88^3.34^Ala exchange was needed to produce stable crystals [81].

These four structures show two distinct allosteric binding sites and motives, which fits very well with reported pharmacological findings. A summarized view of the structural findings about the different allosteric sites at the FFA1 receptor is shown in Figure 2. Partial allosteric agonists could be localized to site A, whereas full allosteric agonists were found to occupy site B. 

Srivastava et al. provided the first crystallographic structure of an FFAR1 ligand complex with the partial agonist TAK-875. Herein, TAK-875 was found to bind to FFAR1, between the transmembrane helices 3–5 and the extracellular loop 2 (ECL2) in a site, henceforth denoted as site A. The ligand bound closer to the exterior membrane surface than most ligands in other GPCRs. TAK-875 seems to enter a noncanonical binding pocket most likely via the lipid layer, resulting in an unusual method of binding. TAK-875 was pharmacologically characterized as an ago-allosteric ligand; thus, this unique binding site most likely represents an allosteric binding pocket in the receptor [201] (Figure 3).

Upon visual inspection of the receptor surface, two extra binding pockets in the vicinity of the TAK-875 site were described [201]. The first pocket is adjacent to TAK-875 and enlarges its binding pocket. It might allow a potential ligand to pass between transmembrane helices 4 and 5, especially as in the human FFA1 receptor the conserved proline 4.60 is replaced by glycine 4.58, which might add some flexibility to the end of transmembrane helix 4. A potential third binding pocket can be found in a location very similar to the binding site of the allosteric modulator LY2119620 in the M2 receptor [203]. So far, the roles of these sites cannot be determined by biological data.

Distinct from the TAK-875 binding site, crystallographic data showed another site in a lipophilic region of the receptor occupied by a monoolein molecule. Lu et al. showed that this binding site (here site B) is involved in binding of full allosteric agonists. They solved the crystallographic structure of the trimeric complex of the human FFAR1 with the partial allosteric agonist MK-8666 and full allosteric agonist AP8. MK-8666 unsurprisingly bound to the TAK-875 binding site (site A), whereas AP8 could surprisingly be found in site B, an extrahelical allosteric, lipid-facing hydrophilic pocket defined by transmembrane helices 3-5 and ICL2 [81].

Ho et al. discovered binding of yet another full agonist of FFAR1 (compound 1) to the second allosteric site (site B). Despite site B being located in a nonpolar region of the receptor, the carboxylic moiety of compound 1 can be accommodated by four hydrogen bond interactions to Tyr44^2.42^, Tyr114^ICL2^, Ser123^4.42^, and one water [84]. In this study, only a binary complex is reported—no additional partial allosteric agonist was involved. 

The second allosteric binding site (site B) was further validated by Lu et al. using site-directed mutagenesis. Mutations of key residues in the site (Y44^2.42^F, Y114^ICL2^F, S123^4.42^A, G95^3.41^F, and A99^3.45^Y) had a detrimental effect on potency [81]. A mutation which showed moderate loss of affinity A102^3.48^W helped explain the loss of activity of AP8 on the dog FFA1 receptor, as this alanine 102 is a valine in the orthologous dog FFA1 receptor. 

By studying crystallographic data of the binary complex of FFAR1 and MK-8666, Lu et al. hinted at structural differences in the arrangement of TM4 and TM5 when compared to the ternary complex of FFAR1–MK-8666–AP8. An interhelical sliding of TM5 in relation to TM4 was observed between the two complex structures. Leu190^5.46^ is translocated by this sliding motion, generating a deep hydrophobic pocket, which can accommodate the CF_3_ moiety of AP8 [81]. This induced fit is apparent as site B is not fully formed in the binary complex without AP8 or in the study of Srivastava et al., despite a monoolein molecule binding at approximately the same position (Figure 4).

The conformation of intracellular loop 2 (ICL2) is another distinct difference between the four published structures. In binary complexes of FFAR1 with the partial allosteric agonists MK-8666 or TAK-875, ICL2 is presumably disordered and not visible. However, if a full allosteric agonist is present in site B, a hydrogen bond from its carboxylate moiety to Tyr114^ICL2^ is formed, which apparently stabilizes ICL2 toward a clearly visible helical structure, thus forming the bottom of the binding site. Without the agonist present, the polar side chain of Tyr114^ICL2^ would not be able to form a hydrogen bond in this hydrophobic region of site B (Figure 5).

In summary, there is detailed knowledge on the overall structure of FFAR1 and especially its allosteric sites, whereas structural considerations about the other members of the FFARs is based only on homology models since crystal structures of these members are currently not available. 

## 7. Opportunities and Challenges of Allosteric GPCR Ligands

Allosteric modulation of bioactive molecules is a natural phenomenon that has been intensely studied and also therapeutically exploited by the drug industry. This is due to several advantages that distinguish allosteric targeting from classical orthosteric drug approaches. 

First, allosteric ligands can promote *complex pharmacology*. The multitude of possible allosteric interaction points for this class of ligands opens the possibility to induce a variety of different receptor conformations entailing distinct signaling outcomes. If ligands can selectively trigger subsets of this signaling repertoire but leave others untouched or inactivate in a given cellular and physiological context, they are referred to as biased ligands (reviewed by [204]). As described above, the capacity to engage various pathways downstream of the FFARs leads to speculation on whether the activation of distinct signaling pathways could be therapeutically exploited as it is known that certain signaling events but not others are associated with distinct physiological outcomes. GPCR signaling is known to be multidimensional—i.e., the signaling outcome depends on multiple and variable associations of input/output parameters. Herein, allosteric ligands could entail distinct pharmacological profiles by fine-tuning GPCR signaling based on their ability to regulate those different parameters. As such, allosteric modulators acting on efficacy (β-factor) by altering the magnitudes of signal response in a quantitative but also qualitative way is a central characteristic of biased ligands (e.g., potentiators or antagonists, inverse agonists, partial or protean agonists, and functionally selective ligands). Combined with the ability to modulate the affinity of the receptor to other binding partners (α-factor), this could generate various phenotypes, such as PAM antagonists, which have been described elsewhere [3]. Modulation can also take place at the level of receptor oligomerization, which could produce distinct pharmacology as compared to monomeric receptor biology, as has also been shown for the FFA receptor family [14]. Signaling bias can also be introduced in a temporal dimension—i.e., ligands that determine when the receptor signals occur. This additional layer of complexity was referred to as temporal bias and has been shown for several GPCR ligands, one of which is the allosteric agonist 4-CMTB for the FFA2 receptor [92,93]. Equally important to the time is the location of GPCR signaling. The capacity of GPCRs to respond to changes in localization, i.e., trafficking along, inside or outside the cell accompanied with distinct signaling outcomes can be described as spatially resolved signaling. This phenomenon ranges from receptor desensitization by internalization to signaling competent intracellular receptor complexes to messaging modules constituted by GPCR-ligand containing extracellular vesicle [100,205,206]. It is worth mentioning that even receptors targeted by only one ligand, e.g., orthosteric agonists, can produce complex pharmacology phenotypes as well [204]. This is because GPCRs commonly interact with either membrane or intracellular adaptor molecules to determine the signaling outcome and thus build an allosteric system themselves consisting of ligand, receptor, and adaptor molecule(s), which is amenable to the aforementioned modulations. The existence of multiple allosteric receptor domains, however, extends the possibilities of how complex GPCR pharmacology can be triggered, modulated, and potentially fine-tuned by ligands targeting these allosteric sites. 

Second, allosteric ligands that do not exert signaling on their own, thus excluding (inverse) allosteric agonists, can create *physiological pharmacology*. That is, they maintain the physiological tone, timing, and location of the natural receptor stimulus. This trait can be advantageous if systemic receptor activation, as it would occur with a high-efficacy orthosteric agonist, is deemed unwanted and only locally released endogenous receptor agonists are potentiated. Another benefit could be that receptor desensitization, as it would occur unselectively by high-efficacious orthosteric agonists, may not occur with allosteric modulators as the timing and location of the natural agonist stimulus remains preserved. 

Third, allosteric modulators promise to show *safer pharmacology* compared to orthosteric drugs. The principle of allosterism is its saturability effect, also referred to as the “ceiling-effect”. Irrespective of the administered dose, allosteric drugs reach limited allosteric effects, be it a positive or allosteric modulation of endogenous stimulus tone [4]. This characteristic could potentially avoid side effects due to relative or absolute overdosing and let the therapeutic window of allosteric modulators be fully exploited.

Fourth, allosteric modulators could employ more *selective pharmacology*. Because allosteric receptor sites are commonly less structurally conserved through lower evolutionary pressure as they do not respond to endogenous ligands, allosteric ligands with unique and more selective binding features can be identified or designed. Several allosteric sites per receptor protein, at least theoretically, further increase the likelihood of finding selective ligands from a drug discovery point of view. De facto selectivity of an allosteric ligand is further increased due to the fact that selective binding with high affinity to the desired receptor subtype is decisive for a selective mode of action, and also that the combination of affinity and efficacy for a certain receptor subtype signaling pathway, reflected in the co-operativity factors α and β, determines whether a ligand produced an actual effect at a given receptor or not. Thus, both requirements must be fulfilled in order to evoke a biological effect, which increases the probability of achieving a selective pharmacological profile with an allosteric drug. 

Hence, it is not surprising that the drug industry invests large amounts of money in programs aiming to discover allosteric drug candidates. While multiple drugs with allosteric modes of action have already reached the market, the proportion of GPCR targeted allosteric ligands is below that of other target groups such as ion channels. This might be due to the pharmacological complexity of this class of receptor proteins and, given that the majority of GPCRs are still deemed orphan and unexplored, this will likely change in the future with increasing insight into GPCR biology. Nevertheless, at least six approved drugs are in use that target GPCRs allosterically, although not all of them have been a priori designed to be allosteric and were later identified to act in an allosteric fashion—e.g., Cinacalcet (Mimpara, Amgen) for the treatment of primary and secondary hyperparathyroidism by increasing the sensitivity of the calcium sensing receptor (CaR) [207]. Ticagrelor (Brilique/Brilinta, AstraZeneca), used to treat acute coronary syndrome, is a negative allosteric modulator at the P2Y_12_ receptor, lowering ADP-induced thrombocyte activation [208]. The antihypertensive and diuretic cyclothiazide (Anhydron, Eli Lilly) acts upon other glutamate and GABA receptors also at the metabotropic mGluR1 receptor as an allosteric antagonist [209]. Maraviroc (Selzentry, Pfizer/GSK) is a negative allosteric modulator of the CCR5 receptor to block entry of HIV into host immune cells such as macrophages or T cells [210]. Plerixafor (Mozobil, Genzyme/Sanofi) is a negative allosteric modulator at the CXCR4 receptor that promotes stem cell release for autologous transplantation [211]. Niclosamide (Bayclusid, Bayer), used as antihelmintic to treat tapeworm infestations, was found to act as a positive allosteric modulator at the neuropeptide Y4 receptor [212].

Despite the aforementioned advantages and examples of successful achievements of targeting GPCRs allosterically by the drug industry, there are major challenges in the discovery and development of allosteric GPCR drugs. 

First, high-throughput screening (HTS) of large compound libraries to find allosteric modulators in functional assays requires the concomitant use of the right coligand. Due to an effect coined “probe dependency”, the allosteric effect of a ligand can change when the nature of the cobinding ligand changes. For instance, the same allosteric ligand can generate a wanted allosteric modulation for an orthosteric compound (compound A), but can show a completely different, potentially unfavorable, pharmacological profile for another orthosteric compound (compound B). Thus, the nature of orthosteric ligands must be taken into account in a screening campaign. In most cases, the natural endogenous orthosteric ligand is the desired one as this is the cobinding partner in the physiological setting. As described earlier, the cellular accessory or coupling proteins of a GPCR also shape the overall allosteric system by functioning as cobinding molecules and thus as potential allosteric modulators. Therefore, the cellular background, i.e., the equipment and accessibility of a cell to receptor adaptor proteins, plays a role in the identification of synthetic allosteric ligands by HTS. A balance between technical feasibility and physiological mimicking needs be found.

Second, structure-based approaches to find allosteric ligands are difficult because physiologically relevant structural information on the GPCR conformation landscapes is still scarce. Although the progress in structural biology, and for complex membrane proteins, such as GPCRs, is remarkable, the hitherto reported structures derive from thermostabilized and constitute artificial receptor constructs. Future advances in this field, e.g., to describe the structural flexibility, will have to show whether this leads to useful information to rationally design GPCR ligands, especially for the structurally less conserved domains of allosteric sites. 

Third, once allosteric compounds are identified, medicinal chemistry programs aim to improve these hits in terms of their chemical, pharmacological, pharmacokinetic, metabolic, and safety parameters. For this, a solid understanding of the structure–activity relationship (SAR) is fundamental to perform a rational optimization approach. The SAR of allosteric ligands, however, was found to be unusually narrow and an insufficient understanding of the relationship often leads to unpredictable outcomes of structural changes in the molecule [211,213,214]. While this is partly associated with the aforementioned lack of detailed structural information on the binding pocket and its embedding into the conformational landscape of physiological GPCR topologies, this situation might improve with the evolving understanding of structural GPCR biology.

Fourth, the lower structural conservation of GPCR allosteric sites bears the problem that receptor orthologs might also change, and species differences in terms of allosteric modulation may occur. This situation can thwart preclinical screening cascades of drug discovery programs and calls for detailed studying of potential differences in the respective species orthologs [211]. Less conserved allosteric sites also increase the probability of polymorphisms within these receptor domains that could potentially change the allosteric communication in the receptor molecule, resulting in a different pharmacological profile. Here, insights into genetics of the targeted patient population can help evaluate the potential risk and can inform preclinical risk mitigation experiments [215]. 

## 8. Conclusions

Allosterism of GPCRs holds the promise to introduce new pharmacology and to unlock traditionally inaccessible modes of actions to eventually treat patients in need. There are a variety of possible mechanisms ranging from positive allosteric modulation to negative allosteric modulation, with all the nuances in between, whether by neutral, partial, inverse agonists, surmountable or insurmountable antagonists, and combinations thereof. Considering the depth of the GPCR biology toolbox, e.g., biased signaling or spatiotemporal control of (sub)cellular or tissue specific signaling, the possibilities awaiting exploration by academic and industry researchers are vast. The first allosteric GPCR modulators that have already reached the market contribute to the necessary learning curve on how this intricate new modality can be medically leveraged. The growing number of academic–industry and public–private partnerships is another manifestation of the recognition that much still needs to be explored in terms of the fundamental understanding of GPCR allosteric biology and of the prospect that transforming basic science into real drugs provides novel opportunities to address unmet medical need. The family of receptors responding to free fatty acids play roles in various processes and lend themselves as promising drug targets. The FFA1 receptor is probably the most studied and medically exploited member of the FFA receptor family. Reports of discontinuation or stagnation of clinical development programs, most prominently of the FFA1 allosteric modulator fasiglifam/TAK-875 from Takedadue to unexpected liver toxicity findings, are certainly discouraging, but seem to be target-independent and compounds have been shown to be amenable to optimization [72,73,74]. The growing number of scientific reports on novel FFAR1 pharmacology based on allosteric modulators, however, sparks renewed interest in the target. 

The next best studied member of the family is the FFA2 receptor. The orthosteric FFA2 antagonist GLPG0974 [216,217] was developed for the treatment of intestinal bowel disease and was found to be safe in phase 1 but nonefficacious in a proof-of-concept phase 2 trial [218]. The lack of efficacy of GLPG0974 needs to be seen in context with the complex pharmacology of FFAR2 in different disease settings. As elaborated earlier, results from genetically modified preclinical models have been conflicting and have triggered a discussion as to whether FFAR2 agonists or antagonists are the preferred approaches [139]. Novel findings from allosteric FFAR2 ligands reveal promising pharmacological traits and other promising results have generated new interest in this target considering novel allosteric modalities. 

GLPG1205 is an antagonist and negative allosteric modulator for GPR84 [195] that was discontinued for the development of IBD due to the negative ORIGIN phase 2 trial (NCT02337608) but showed promising preclinical results in models of idiopathic pulmonary fibrosis [191] and also positive topline results from the proof-of-concept PINTA phase 2 study in IPF patients (NCT03725852). 

Hence, allosteric targeting of receptors for small-, medium-, and long-chain fatty acids is a promising approach to address patient needs in different therapeutic areas mirroring the biological versatility of the receptor family members and their attractiveness as drug targets. 

## Figures and Tables

**Figure 1 ijms-22-01763-f001:**
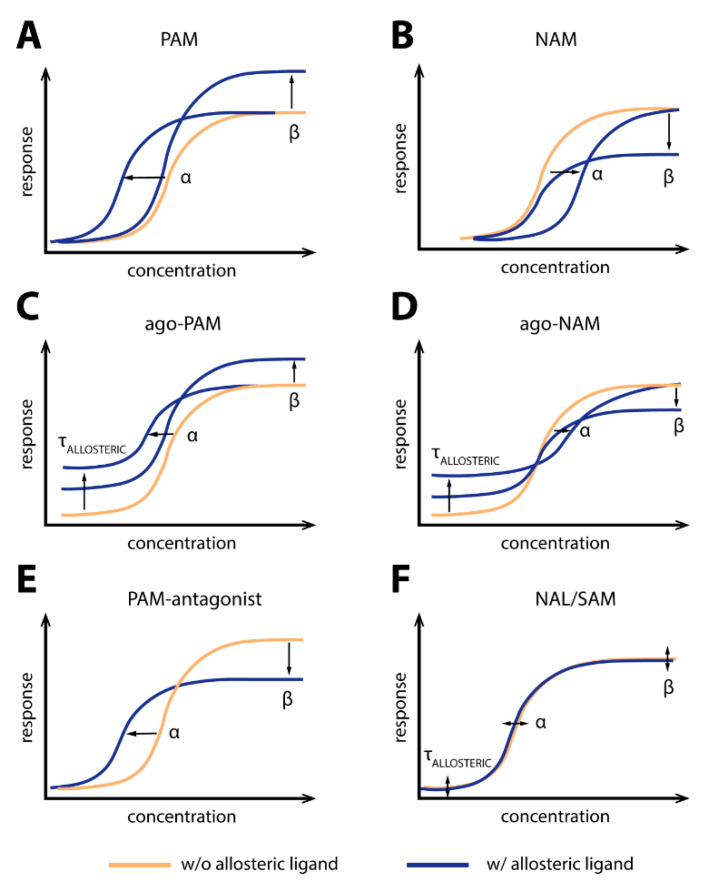
Overview of different modes of allosteric modulation. Agonist dose–response curves in absence of an allosteric ligand are shown in orange, while presence of an allosteric ligand is depicted in blue. (**A**) A positive allosteric modulator (PAM) increases affinity and/or efficacy of an orthosteric ligand, and thus has positive co-operativity factors α and/or β. (**B**) A negative allosteric modulator (NAM) reduces affinity and/or efficacy of an orthosteric ligand and has a negative α- and/or β-factor. (**C**) An ago-PAM has intrinsic efficacy (τ > 0) and is also a PAM for an orthosteric ligand. (**D**) An ago-NAM is a negative allosteric modulator for an orthosteric ligand that increases the activity state of the receptor (τ > 0) itself. (**E**) A PAM-antagonist decreases efficacy of an orthosteric agonist and thus functions as an antagonist (β < 0) and simultaneously increases affinity of the orthosteric ligand (α > 0). (**F**) A negative allosteric ligand (NAL) or silent allosteric modulator (SAM) has no effect on the affinity or efficacy of an orthosteric ligand (α,β = 0) but occupies the allosteric binding site, and thus competes with other allosteric ligands for this site.

**Figure 2 ijms-22-01763-f002:**
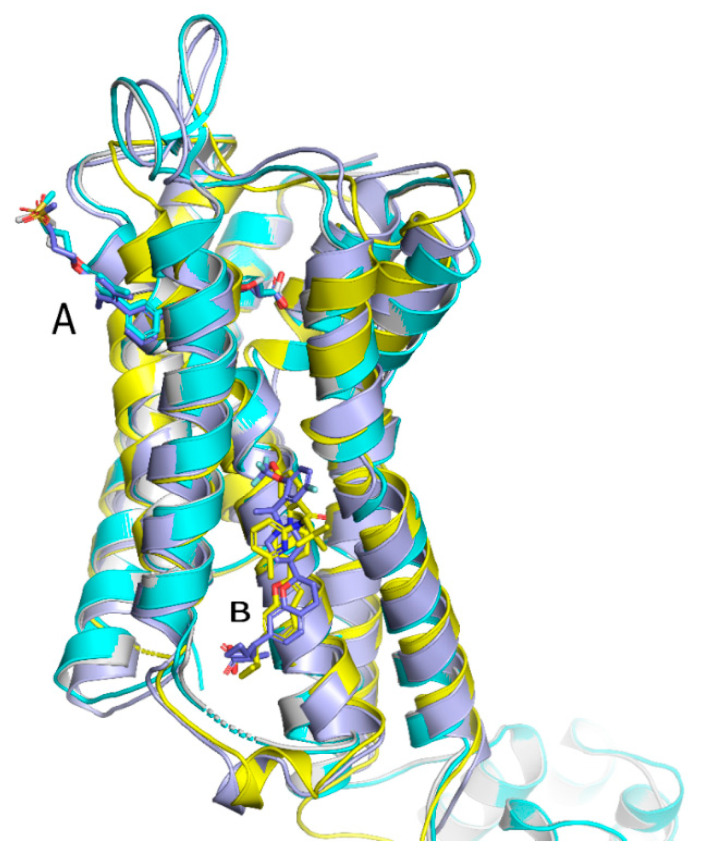
Superimposition of all published FFAR1 crystallographic structures. The partial allosteric agonists TAK-875 and MK-8666 share the same binding site, denoted here as site A. Full allosteric agonists (AP8 and compound 1) are found in the second allosteric site, denoted here as site B. The respective PDB IDs are: 4PHU, 5TZY, 5TZR, and 5KW2. This figure was prepared using The PyMOL Molecular Graphics System, Version 2.0 Schrödinger, LLC (New York, NY, USA).

**Figure 3 ijms-22-01763-f003:**
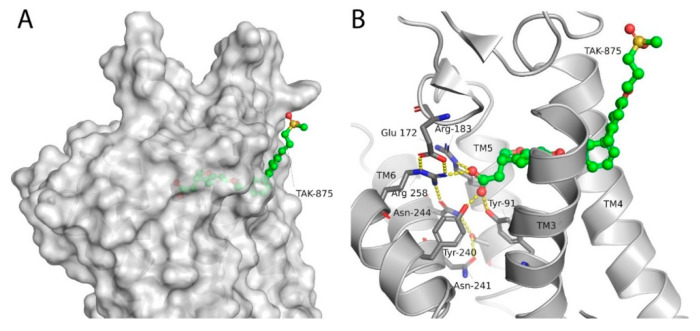
(**A**) The partial agonist TAK-875 seems to enter its allosteric binding pocket (site A) in an unusual way, most likely through the lipid layer, sneaking into the receptor transmembrane region. (**B**) Detailed view on binding pose of TAK-875 with the head group buried in the receptor transmembrane region. Hydrogen bonding pattern of TAK-875 with key residues in its allosteric binding pocket shown as dashed lines. This figure was prepared using The PyMOL Molecular Graphics System, Version 2.0 Schrödinger, LLC (New York, NY, USA).

**Figure 4 ijms-22-01763-f004:**
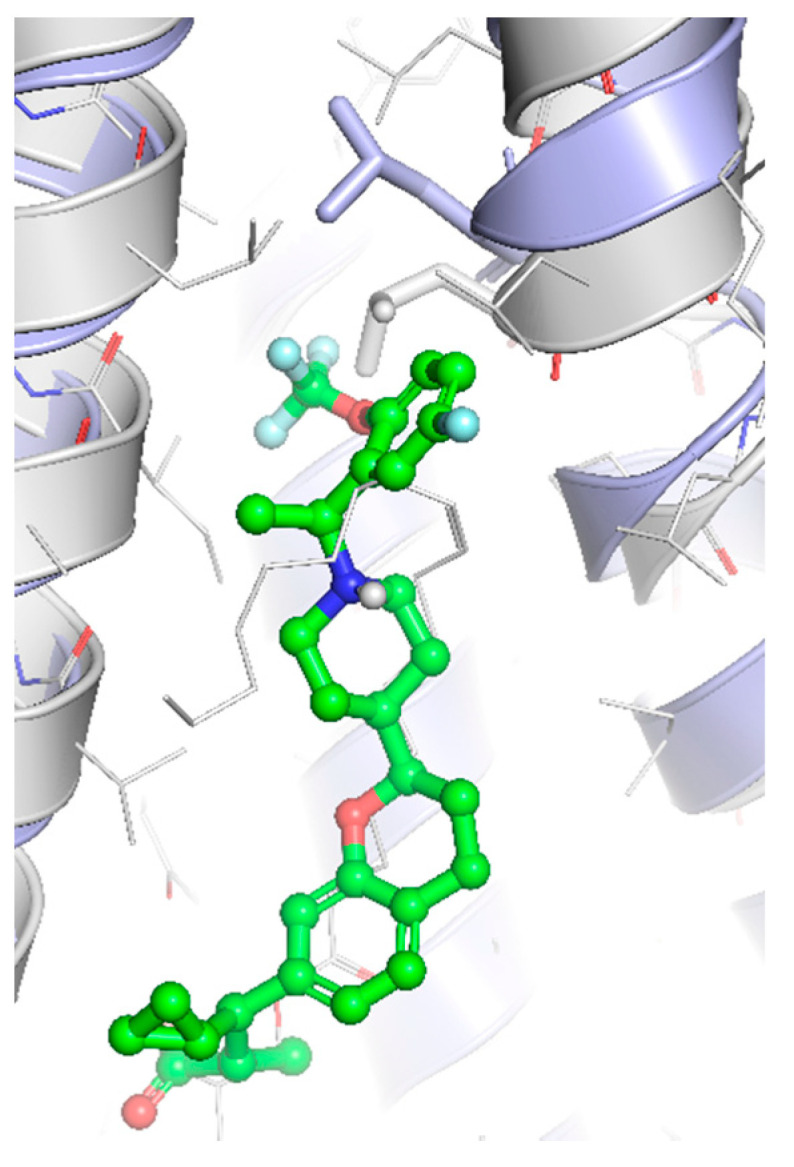
An interhelical sliding of TM5 in relation to TM4 leads to a translocation of Leu190^5.46^. This generates a deep hydrophobic pocket to accommodate the CF3 moiety of the full allosteric agonist AP8 binding to the second allosteric binding pocket (site B). Binary complex of FFAR1 and MK-8666 (PDB ID: 5TZR) shown in grey and ternary complex, consisting of FFAR1, MK-8666, and AP8, (PDB ID: 5TZY) is depicted in blue. This figure was prepared using The PyMOL Molecular Graphics System, Version 2.0 Schrödinger, LLC (New York, NY, USA).

**Figure 5 ijms-22-01763-f005:**
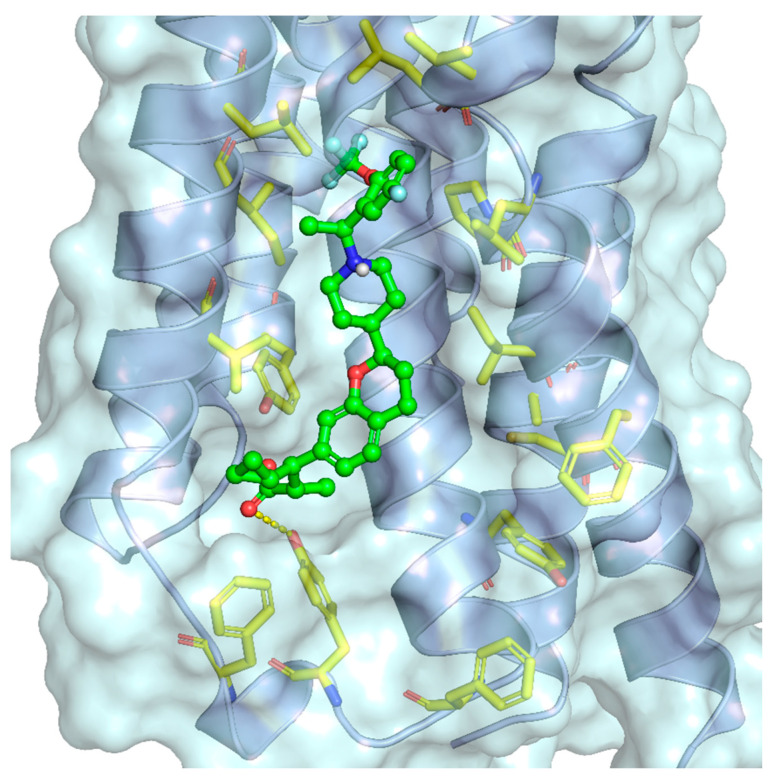
The intracellular loop 2 (ICL2) of FFAR1 at the bottom of the receptor–ligand model is stabilized via a hydrogen bond (dashed line) of full allosteric agonist AP8 to Tyr114. This figure was prepared using The PyMOL Molecular Graphics System, Version 2.0 Schrödinger, LLC.

**Table 1 ijms-22-01763-t001:** Summary of free fatty acid (FFA) receptor expression profile and biological functions.

Receptor	Major Coupling Proteins	Major Expression Sites	Main Function
FFAR1	Gαq/11 Gαi/o Gαs Gα12/13 β-arrestin	Pancreas (β-cells) Intestine (L, K, I cells) Bone Central nervous system Immune cells (Monocytes)	Insulin secretion Gut hormone secretion Bone remodeling Pain perception Macrophage M2 differentiation
FFAR2	Gαq/11 Gαi/o Gα12/13 β-arrestin	PMNs (Neutrophils, Eosinophils) Lymphocytes Monocytes Pancreas (β-cells) Intestine (L cells, IECs) White adipose tissue	Immune cell activation T_reg_ expansion Cytokine secretion Insulin release Gut hormone secretion, immune-modulatory Reduction in lipolysis, lipid accumulation, and insulin resistance
FFAR3	Gαi/o β-arrestin	Peripheral nervous system Pancreas (β-cell) Intestine (L, K cells) Immune tissue (DCs, thymus)	Increase in heart rate, energy expenditure, reduction of gut motility Inhibition of insulin secretion Gut hormone release Decrease Th2 response, increase T_reg_ differentiation
FFAR4	Gαq/11 Gαi/o β-arrestin	Adipose tissue Macrophages Lung Intestine (K, I cells) Bone	Differentiation, browning Anti-inflammatory Epithelial repair Gut hormone release Bone formation
GPR84	Gαi/o β-arrestin	Immune cells (Neutrophils, Eosinophils, macrophages) Lung, Liver, Muscle, and Adipose tissues	Proinflammatory cell responses

**Table 2 ijms-22-01763-t002:** Selection of Free Fatty Acid Receptor 1 (FFAR1) allosteric ligands.

Name	Structure	References
**Partial allosteric agonists**		
TAK-875/fasiglifam	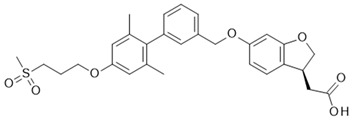	Takeda Pharmaceuticals [70,71,78]
AM 837	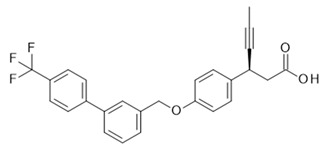	Amgen [62,79]
MK8666	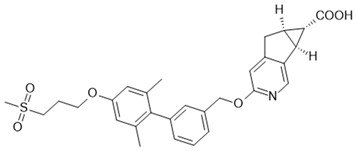	Merck [68,80,81,82]
**Full allosteric agonists**		
AM 1638	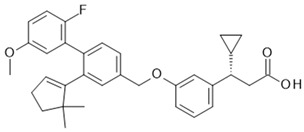	Amgen [62,83]
AP8	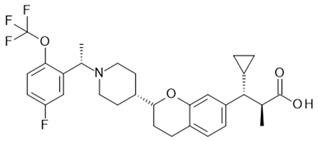	Merck [81]
Compound 1	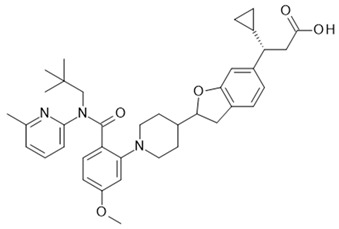	Eli Lilly [84]

**Table 3 ijms-22-01763-t003:** Selection of FFAR2 and FFAR3 allosteric ligands.

Name	Structure	References
**FFA2 allosteric agonists**		
AMG 7703/4-CMTB	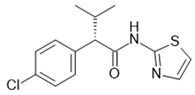	Amgen [88,91,92,105,107]
Compound 58	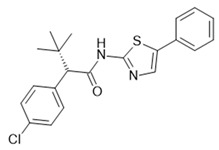	Amgen [89,141]
AZ1729	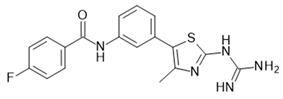	AstraZeneca [145]
**FFA3 allosteric agonist**		
AR420626	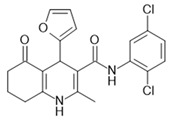	Arena Pharmaceuticals [146,147]

**Table 4 ijms-22-01763-t004:** FFAR4 allosteric ligand.

Name	Structure	References
**Negative Allosteric Modulator**		
AH-7614	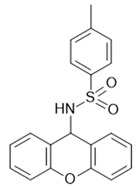	GlaxoSmithKline [30,169]

**Table 5 ijms-22-01763-t005:** GPR84 allosteric ligands.

Name	Structure	References
**Allosteric Agonist**		
DIM	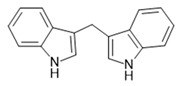	[180,181]
**Negative allosteric modulator**		
GLPG1205	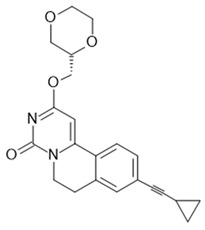	Galapagos NV [195]

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
