# Peer review of "Pharmacology of Free Fatty Acid Receptors and Their Allosteric Modulators"

_ijms, 2021, doi:10.3390/ijms22041763_

Round 1

Reviewer 1 Report

This very rich and up-to-date review by M. Grundmann et al. deals with the free fatty acid receptors (FFAR). It also offers a beneficial and welcome introduction concerning the whole family of GPCRs to which FFARs belong.

General comment :

The comprehensive panorama of FFARs’ allosteric ligands is of high interest as well as the discussions concerning the mechanisms involved in functional interactions between orthosteric and allosteric ligands in the different FFARs. My only concern is that only intramolecular mechanisms are considered. Since many (most ?) other GPCRs act through dimerization or multimerization, it would certainly be of interest to consider the possibility (even if never demonstrated until now) that « allosteric » ligands can act through homologous or heterologous interactions of FFARs.

Specific comments :

  • Figure 2 is not very useful. Its information (G protein(s) interacting with each FFAR) could be included in Table1 instead.
  • Table 1 could also include the names of orthosteric ligands for each FFAR : FFAR1 (LCFA), FFAR2 (SCFA), FFAR3 (SCFA), FFAR4 (LCFA), GPR84 (MCFA).
  • The sentence in lines 513-517 is not very clear to me.

Reviewer 2 Report

This review, entitled “Pharmacology of free fatty acid receptors and their allosteric modulators,” authored by Grundmann et al., reports valuable information to understand the pharmacology and allosteric modulation in FFARs. The surveys represent a notable advance in the development of knowledge about the pharmacology and allosteric modulation of FFARs. The historic finding to connect recent work is excellent. In my opinion, this is a useful review, and the manuscript is suitable for publication in the journal after the authors have addressed the following comments and questions:

Major questions:

1) How right the in-silico approach is of finding allosteric sites to modulate the activity in FFARs? In addition, homology modeling of FFARs other than FFAR1 will create an additional round of shade in the precise prediction of these sites. Even co-crystallization of FFAR1 with an allosteric modulator may show non-specific interactions because of the compounds' hydrophobic nature, which are typically used in high concentrations during soaking.

2) Do you have any docking study and/or mutational analysis performed for the FFARs to give a clear picture of allostery? Superimposing crystal structure with a homology model to track out the allosteric site is straightforward (Figure 3 and 5).

3) Is there any possibility of allosteric modulators binding the orthosteric sites too?

Minor questions/suggestion:

1) In figure 1, please specify the coloring of traces (like intrinsic has orange and modulators has blue). It will make the review better to understand. Also, please describe in the legend or somewhere…what is ago-PTM or any abbreviated forms, at least first time used, it will make easy for readers of non-pharmacology background.

2) some corrections…

Line 133…. signaling seem to be…..> signaling seems to be  

Line 339…. agonists reduces food …..> agonists reduce food

Line 147…. The most advance FFA1…..> The most advanced FFA1  

Line 899…. FFARs is of mayor importance…..> FFARs is of major importance or FFARs is of significant importance

Line 921…. deducted that theses mutations …..> deducted that these mutations

Line 1017…. advantages that that distinguish …..> advantages that distinguish
